# Representational untangling by the firing rate nonlinearity in V1 simple cells

Merse E Gáspár[1,2], Pierre-Olivier Polack[3], Peyman Golshani[4,5,6], Máté Lengyel[2,7], Gergő Orbán[1]*

[1]MTA Wigner Research Center for Physics, Budapest, Hungary; [2]Department of Cognitive Science, Central European University, Budapest, Hungary; [3]Center for Molecular and Behavioral Neuroscience, Rutgers University, Newark, United States; [4]Integrative Center for Learning and Memory, Brain Research Institute, University of California, Los Angeles, Los Angeles, United States; [5]Department of Neurology, David Geffen School of Medicine, University of California, Los Angeles, Los Angeles, United States; [6]West Los Angeles VA Medical Center, Los Angeles, United States; [7]Department of Engineering, Computational and Biological Learning Lab, University of Cambridge, Cambridge, United Kingdom

**Abstract** An important computational goal of the visual system is 'representational untangling' (RU): representing increasingly complex features of visual scenes in an easily decodable format. RU is typically assumed to be achieved in high-level visual cortices via several stages of cortical processing. Here we show, using a canonical population coding model, that RU of low-level orientation information is already performed at the first cortical stage of visual processing, but not before that, by a fundamental cellular-level property: the thresholded firing rate nonlinearity of simple cells in the primary visual cortex (V1). We identified specific, experimentally measurable parameters that determined the optimal firing threshold for RU and found that the thresholds of V1 simple cells extracted from in vivo recordings in awake behaving mice were near optimal. These results suggest that information re-formatting, rather than maximisation, may already be a relevant computational goal for the early visual system.

DOI: https://doi.org/10.7554/eLife.43625.001

**\*For correspondence:**
orban.gergo@wigner.mta.hu

**Competing interests:** The authors declare that no competing interests exist.

## Introduction

The visual cortex relies on a series of hierarchically organised processing stages to construct increasingly complex representations of the visual environment (*Orban, 2008*; *Tafazoli et al., 2017*; *Ungerleider and Haxby, 1994*). An important goal of this processing hierarchy in the ventral visual stream is to represent information about stimuli in a format that facilitates behaviourally relevant tasks, such as object recognition, identification, or classification. This reformatting of information has been called 'representational untangling' (RU; *DiCarlo and Cox, 2007*; *Bengio et al., 2013*) and it is often formalised via the concept of linear decodability. Linear decoding measures the extent to which a particular stimulus feature (such as the identity of an object) is explicitly encoded in a representation such that it can be accurately estimated based on a simple weighted sum of the activation of representational units (*Bishop, 2006*). For example, classification boundaries for different objects appear highly non-linear and 'tangled' in the space of pixel intensities, or the space of retinal or primary visual cortical (V1) activities, thus preventing efficient linear decoding of object identity information. In contrast, in inferior temporal cortex (IT), these boundaries become untangled, making object identity information linearly decodable (*DiCarlo et al., 2012*). Critically, linear decodability of a stimulus feature does not only require that neural responses are modulated by this feature, but

also that at the same time they remain tolerant to, that is depend only weakly or trivially on other, 'nuisance' stimulus features that can also change across stimuli. For example, while neural responses across the whole visual system, from the retina and V1 to IT, will change when different objects are presented, the activation of select IT cells shows tolerance to changes in 'nuisance' parameters, such as illumination, location or scale (*Brincat and Connor, 2004*; *DiCarlo and Cox, 2007*; *Ito et al., 1995*; *Logothetis et al., 1994*; *Tanaka, 1996*; *Vogels and Biederman, 2002*).

The neural mechanisms underlying RU are largely unknown, with most previous work focussing on RU of object category information in IT, and how the cascaded nonlinear input-output transformations of the early stages of the visual hierarchy contribute to it (*Hung et al., 2005*; *Pagan et al., 2013*; *Yamins and DiCarlo, 2016*). In contrast, we study an elementary form of RU that is already taking place at the first stage of visual cortical processing, in V1, and uses a simple and ubiquitous property of single neurons: the firing rate nonlinearity (FRNL), that is the nonlinear transformation between a cell's membrane potential and its instantaneous firing rate. In V1, there is broad agreement that an image feature that is explicitly represented is local orientation. Indeed, several studies investigated linear decodability of stimulus orientation directly from V1 firing rates (or spike counts) as a function of tuning curve properties (*Ecker et al., 2011*; *Seriès et al., 2004*; *Seung and Sompolinsky, 1993*; *Shamir and Sompolinsky, 2006*), noise correlations (*Berens et al., 2012*; *Moreno-Bote et al., 2014*), or the internal dynamics of V1 (*Gutnisky et al., 2017*). However, previous work did not examine membrane potential responses, and was thus not suitable for studying the specific contribution of the FRNL to linear decodability. Moreover, these studies only considered at most a single nuisance parameter (contrast), thus requiring only minimal RU to be carried out in neural responses.

In order to study the contribution of the FRNL to RU in V1, we directly compared the linear decodability of stimulus orientation from membrane potentials and firing rates, and considered some of the most prevalent nuisance parameters of visual stimuli: contrast (an elementary aspect of illumination), phase (location), and spatial period (scale). We found that nuisance parameters made the linear decodability of orientation information non-trivial: once this richer set of nuisance parameters was considered, membrane potential responses of a population of orientation-selective cells remained highly tangled, and linearly undecodable. However, despite the obvious loss of *total* information caused by the rectifying aspect of the FRNL, which is due to all membrane potential values below the firing threshold being mapped to zero firing rate, the *format* of information in firing rates was more amenable to the linear decoding of orientation. This trade-off between total information and linear decodability resulted in a clear optimum for the value of the firing threshold. In particular, the optimal firing threshold depended on a few key experimentally measurable parameters, and we confirmed in *in vivo* intracellular recordings that mouse V1 simple cells had their thresholds near the optimal value. These results suggest that RU may be a universal principle of organisation throughout the visual system, and it involves cellular as well as circuit-level mechanisms.

## Results

In order to study the RU of information about stimulus orientation in V1 responses, we adapted a canonical population coding model (*Jones and Palmer, 1987*) (*Figure 1*). We chose this model as it had the minimal complexity necessary for systematically studying the effects of specific single-neuron (e.g. FRNL threshold, membrane potential variability) and network parameters (e.g. population size, noise correlations) on the encoding of stimulus orientation in the face of other (nuisance) stimulus features affecting neural responses. More specifically, the membrane potential response of each model neuron was determined by the linear response of the neuron-specific oriented Gabor filter to the stimulus. These Gabor filters approximated the receptive field properties of simple cells, resulting in response characteristics that were modulated by the orientation, frequency, and phase of a static, full field sinusoidal grating stimulus (*Dayan and Abbott, 2005*). To model the *in vivo* variability of V1 membrane potential responses in awake animals (*Haider et al., 2013*), these membrane potentials were then subject to additive Gaussian noise independently sampled in time windows whose length (20 ms) was approximately matched to the decay time of the autocorrelation function of simple cells (*Azouz and Gray, 1999*). The firing rate of each cell was obtained using a rectifying nonlinearity that has been shown to capture the FRNL of simple cells (*Carandini and Ferster, 2000*; *Dorn and Ringach, 2003*; *Priebe and Ferster, 2008*). RU was quantified by the performance of a

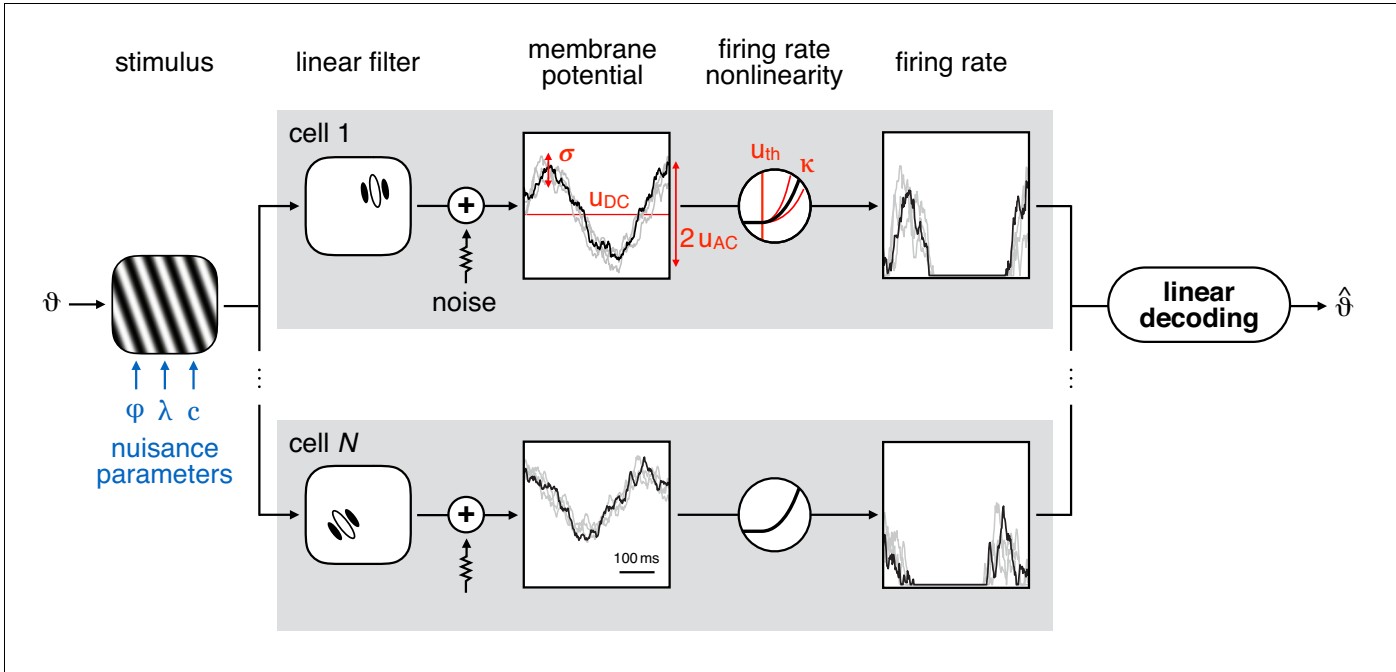

**Figure 1.** Model schematic. Left: sine wave grating stimulus used as input to the model population. The stimulus is parametrised by orientation ($\vartheta$) and nuisance parameters phase ($\varphi$), spatial period ($\lambda$), and contrast ($c$). Middle: simple cell membrane potential (MP) responses (grey boxes) are obtained using localised, oriented Gabor filters and temporally correlated additive noise (Gaussian, with standard deviation $\sigma$). The mean response varies sinusoidally (with amplitude $u_{AC}$) around the baseline ($u_{DC}$) as the phase of the grating stimulus is changing. The stochastic component of the MP response of each neuron is variable in time (black line) and across trials (gray lines). Firing rate is obtained by transforming the MP through a threshold-power-law firing rate nonlinearity (FRNL), characterised by threshold $u_{th}$ and exponent $\kappa$. Right: linear decoding of orientation from the population response of $N$ simple cells yields the estimated orientation ($\hat{\vartheta}$).

DOI: https://doi.org/10.7554/eLife.43625.002

linear decoder which decoded stimulus orientation from membrane potentials or firing rates in the face of noise and variability in other (nuisance) parameters of the stimulus: phase, contrast, and spatial frequency (*Figure 1*, blue).

We studied RU as a function of parameters describing the stimulus distribution as well as parameters describing the neural population. As the complete parameter space of the model (including the detailed stimulus filter of every neuron) was vast, it was unfeasible to explore it fully. Thus, we focussed on a few key characteristics of our model neurons (*Figure 1*, red): the mean membrane potential ($u_{DC}$) and depth of modulation ($u_{AC}$), which were defined based on the membrane potential response to a drifting full-field grating at 100% contrast, preferred orientation and preferred spatial period; noise variability ($\sigma$), which determined the magnitude of the noise injected into membrane potentials; and the threshold ($u_{th}$) and exponent ($\kappa$) of the single neuron FRNL. At the population level, we studied the effects of population size ($N$); decoding resolution ($K$), defined as the number of different orientation categories to be decoded; and the magnitude of noise correlations ($\rho$) with a given structure, that is the correlation between the membrane potential noise of different cells in the population.

## The effect of response rectification on representational untangling

The effects of noise and nuisance parameters on linear decodability can be understood by considering the binary discrimination of two orientations in a pair of neurons responding to stimuli with variable orientation (to be decoded) and phase (to which the decoder is required to be invariant) (*Figure 2*). This binary classification task (discrimination) generalises for multiclass cases that can be regarded as combinations of pairwise comparisons. In the model, filter responses depend both on the orientation and phase of the stimulus: at any particular orientation, variability in phase induces a manifold of responses (*Figure 2A*, coloured ellipses). Membrane potential responses are derived

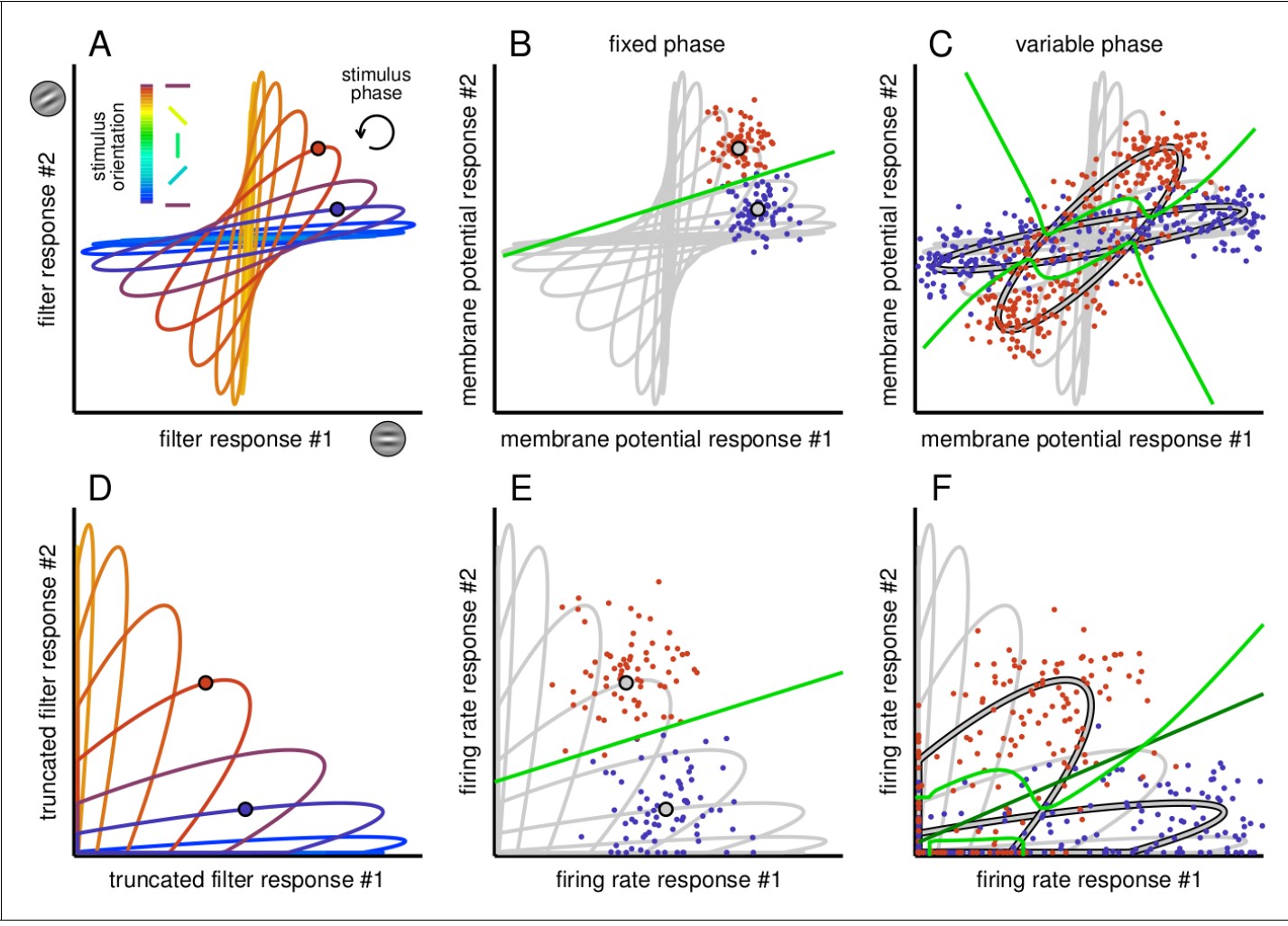

**Figure 2.** Effect of phase variability on the linear separability of the responses of a pair of simple cells. (A) Filter responses of a pair of model simple cells to gratings with various orientations (different colours) and phases (position along ellipses). Circles mark a pair of example stimulus phases at two different orientations. (B) Stochastic membrane potential responses of the pair of cells (dots) at the reference pair of stimulus orientations and phases (gray circles with black contour). Gray lines show average responses at other phases and orientations for reference. The optimal decision boundary (green line) constructed for membrane potential responses is a straight line and it is thus achievable by a linear decoder. (C) Membrane potential responses (coloured dots) to stimuli with the same pair of orientations as in B but for variable stimulus phase (gray ellipses with black contour). Phase variability abolishes the linear separability of responses given to different orientations: the decision boundary of the optimal decoder (green line) is highly nonlinear. (D) Filter responses as in A but truncated due to the threshold linear transformation of the FRNL. (E–F) Same as B-C, but for firing rate responses. Note that the decision boundary of the optimal decoder of firing rate responses (F, green line) can be well approximated by a linear decoder (F, dark green line).

DOI: https://doi.org/10.7554/eLife.43625.003

from filter responses but are contaminated by noise (*Figure 2A*, coloured dots), which introduces uncertainty even when phase is fixed and known. Nevertheless, as long as the phase of the stimulus is fixed (and the noise is not overwhelmingly large), all membrane potential values scatter around the same value for each orientation, creating well separated sets of joint responses so that orientation remains linearly decodable (*Figure 2B*, green line shows optimal classification boundary). However, variability in phase introduces a substantial amount of additional variability in responses along the corresponding manifolds which intersect multiple times. This causes the sets of membrane potential responses to become strongly overlapping (*Figure 2C*, coloured dots) and the optimal decision boundary to become highly nonlinear ('entangled', *Figure 2C*, green line), such that no linear decision boundary can approximate it efficiently. Thus, even the representation of orientation

information by orientation-tuned cells can become highly entangled in the presence of nuisance parameters.

Nonlinear transformations of variables can render even complex representations linearly decodable – an insight that underlies many pattern recognition algorithms (*Bishop, 2006*). Specifically, in our case, we focus on the rectifying aspect of the firing rate nonlinearity of neurons. This rectification effectively 'removes' a large part of the membrane potential response space thus letting the decoder operate only on the quadrant of super-threshold responses (*Figure 2D*). While this drastic removal of a large fraction of responses can clearly lead to severe total information loss (*Barak et al., 2013*), responses in the remaining quadrant may also become more linearly separable. This is because the density of manifold intersections generally decreases towards higher membrane potential values. In other words, the rectification only allows strongly responding cells to contribute to decoding, and the resulting sparsification of the representation will generally render it more linearly separable (*Barak et al., 2013*). This explains why the optimal decision boundary for firing rates remains well approximated by a line (*Figure 2E*), even with variability in stimulus phase (*Figure 2F*, the decision boundary of the optimal decoder is shown in light green while that of the best linear decoder in dark green). Thus, there is a trade-off for RU between total information loss and sparsification controlled by the firing threshold, and this trade-off becomes particularly acute in the face of nuisance parameter variations.

## Orientation decoding from a population under phase variability

In order to study the trade-off between total information loss and sparsification quantitatively, we parametrically varied the firing threshold in a population of $N = 500$ neurons, each characterised by identical firing thresholds and noise but random receptive field parameters (*Figure 3—figure supplement 1A*). To simplify our analysis, we assumed a rectified linear FRNL (see below for FRNLs with exponents greater than one). At each threshold level, we compared the performance of two decoders: a linear decoder, and an optimal Bayesian decoder (see Materials and methods). The linear decoder was trained and tested on different subsets of data differing in the membrane potential noise added to the linear filter responses of model neurons. The training data set was sufficiently large that asymptotic test performance was achieved, therefore the performance of the linear decoder was limited solely by the properties of the stimuli and not by the amount of data. The optimal decoder was constructed with perfect knowledge of the process generating neural responses and thus did not need to be separately trained. As it used the optimal decision boundaries, the performance of the Bayesian decoder represented a theoretical upper bound on the performance of *any* decoder, and could also be used to quantify the total information content of responses (Materials and methods; *Panzeri et al., 1999*). Each decoder was tested with the phase of the stimulus kept fixed (*Figure 3A*, black: linear decoder, gray: optimal decoder) or being varied (*Figure 3A*, dark blue: linear decoder, light blue: optimal decoder).

The performance of the optimal decoder simply decreased monotonically as the threshold was increased (*Figure 3A*, gray and light blue). This was expected because the thresholding effect of the FRNL loses information in the subthreshold range (as in this range all membrane potential values are mapped to the same zero firing rate) while the super-threshold part of the FRNL (even if it is nonlinear) represents a one-to-one mapping which does not change information content. This means that the net effect of the FRNL can only be information loss, with higher thresholds leading to larger information loss (*Figure 3—figure supplement 2B*). Therefore, as long as the functional objective of V1 was the maximisation of total orientation information transmitted to downstream areas (the so-called 'infomax' principle; *Linsker, 1988*; *Bell and Sejnowski, 1997*), one would expect to see low values of the firing threshold, clipping the membrane potential distribution as little as possible (*Figure 3A*, green shaded area).

In contrast to the simple monotonic decrease in total information, linear decoding performance showed a more complex, non-monotonic dependence on the firing threshold. At the lowest values of the threshold, all membrane potential responses were super-threshold (*Figure 3A*, green histogram shows the distribution of membrane potential responses across all stimuli), and so decoding from firing rates was essentially equivalent to decoding from membrane potentials. Thus, as expected (*Figure 2*), linear decoding with variable stimulus phase was at chance (10%) at this extreme, that is it was unable to extract any information about orientation from the membrane potential responses of the population (*Figure 3B*, left blue bar). Correspondingly, the coefficients of

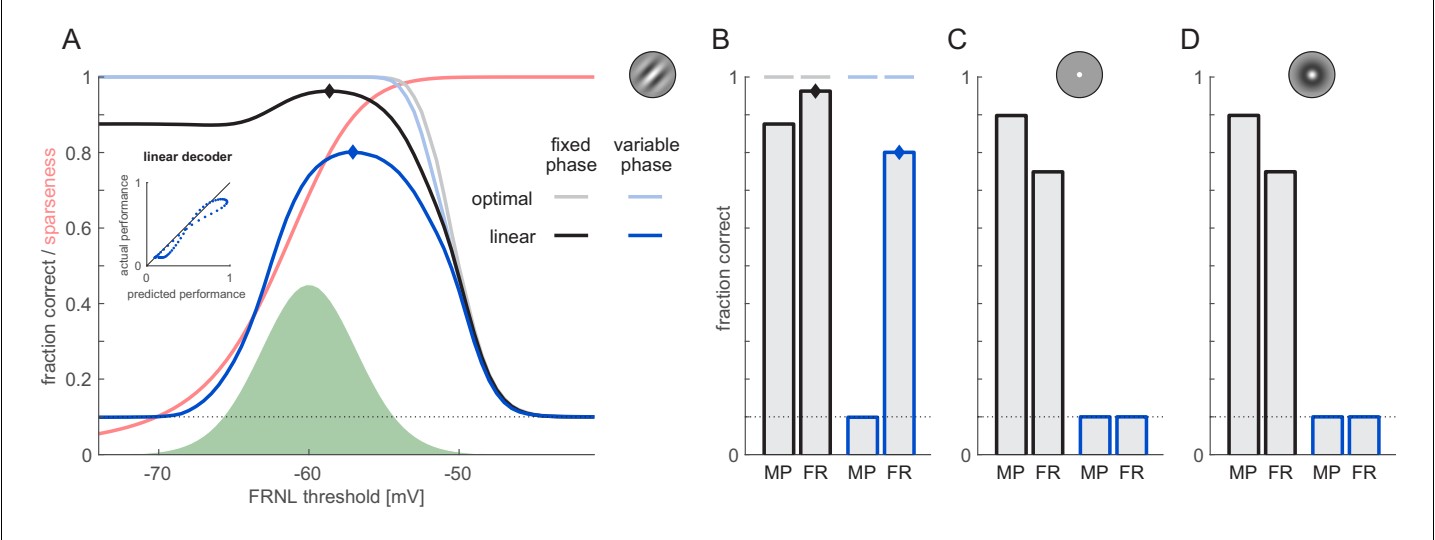

**Figure 3.** Effect of phase variability on decoding performance. Orientation decoding from firing rates (FR) was performed for grating stimuli with fixed (black and grey) or unknown phase (dark and light blue) using an optimal (lighter colours) or linear decoder (darker colours). (A) Decoding performance as a function of the FRNL threshold (solid lines). Black dotted line shows chance performance. Green shaded area shows the distribution of membrane potentials for reference. Red line shows sparseness of responses as a function of the FRNL threshold. Inset shows the performance of the linear decoder against the performance predicted by the normalised optimal decoder (the combined effect of total information and sparseness) at different values of the FRNL threshold under variable phase (blue dots). Note that decoding from firing rates with FRNL threshold values below the membrane potential distribution is equivalent to decoding from membrane potentials. (B) Performance of the linear decoder using firing rates (FR) obtained with the optimal FRNL thresholds (diamonds on A) and membrane potentials (MP). Horizontal lines show the performance of the optimal decoder at the corresponding FRNL threshold values. (C) Performance of linear decoder with fixed (black bars) and unknown phase (blue bars) for a neuron population with pixel-like receptive fields. As no optimal threshold existed with this population, the firing rate decoder was evaluated at the optimal FRNL threshold of the Gabor population. (D) Same as panel C but for a population with center-surround receptive fields.

DOI: https://doi.org/10.7554/eLife.43625.004

The following figure supplements are available for figure 3:

**Figure supplement 1.** Properties of the model neuron population.
DOI: https://doi.org/10.7554/eLife.43625.005
**Figure supplement 2.** Alternative measures for the characterisation of decoding performance.
DOI: https://doi.org/10.7554/eLife.43625.006
**Figure supplement 3.** Linear decoding performance with variable phase for narrow and wide field of view.
DOI: https://doi.org/10.7554/eLife.43625.007

the linear decoder did not bear any systematic relationship with the preferred orientations of neurons and the decoded orientations (*Figure 3—figure supplement 1B*). The failure of linear decoding was due to membrane potential responses fully reverting for stimuli with anti-preferred phases. This meant that depending on stimulus phase, responses for the preferred orientation of a cell could be well above or below responses to non-preferred orientations, thus violating the monotonic relationship that linear decoding requires between responses and the match of stimulus orientation to the preferred orientation of cells. For fixed stimulus phase, linear decodability was well above chance (~87%) even at low threshold values (*Figure 3B*, left black bar). Note that, unlike the case suggested in *Figure 2*, it did not reach the performance of the optimal decoder because we allowed orientation itself to vary within each $K = 10$ discrete decoded orientation categories, making the optimal decision boundaries slightly nonlinear even at a fixed stimulus phase.

At extremely high values of the firing threshold, membrane potential responses always remained sub-threshold, keeping firing rates zero at all times. Thus, all decoders performed at chance due to this total loss of information. Between the two extremes, as the threshold increased, there was a trade-off between two opposing effects: the total information in responses decreased, as shown by the monotonically decreasing performance of the optimal decoder (*Figure 3A*, gray and light blue; see also above), while responses became increasingly sparser (see Materials and methods, *Figure 3A*, red), increasing the linear decodability of the remaining information (see *Figure 2F*). As

a result of this trade-off, linear decoding had a pronounced peak with an optimum at intermediate firing thresholds values, around $V_m = -57$ mV in this case (*Figure 3A*, dark blue diamond; *Figure 3B* right blue bar), that is ~3 mV above the average membrane potential (*Figure 3A*). This optimal firing threshold was largely independent of the precise measure used to quantify performance, whether it was simply the fraction of correct responses used here and in the following, a statistically more appropriate probabilistic fraction correct measure, or a measure that also depended on the magnitude of the (circular) error between true and decoded orientation (*Figure 3—figure supplement 2A*). The success of linear decoding at the optimal threshold was also reflected in the patterns of decoding coefficients: as expected, they scaled with the difference between a neuron's preferred orientation and the decoded orientation (*Figure 3—figure supplement 1C*). To assess the extent to which the FRNL threshold threshold was effective in helping RU using local image patches instead of grating stimuli covering a large portion of the visual field, we also performed the same analysis with a 500-neuron population resembling a V1 hypercolumn, with receptive fields covering only a 3° circle, produced results similar to those obtained with full-field stimuli. (*Figure 3—figure supplement 3*).

In order to see how much the trade-off between total information loss and population sparseness could account for linear decodability, we computed the correlation between the actual performance of the linear decoder and the performance that could be predicted based on scaling the performance of the optimal decoder (FC$_{opt}$, indicative of total information, see above) by population sparseness: (FC$_{opt}$ - chance) $\times$ sparseness + chance. We found a strong correlation across different values of the firing threshold between the actual and predicted performance of the linear decoder ($r = 0.98$, *Figure 3A*, inset). For fixed stimulus phase, decision boundaries were generally more linear, thus the reduction of overall information dominated, which resulted in only a smaller peak in performance at around the same threshold value (*Figure 3A*, black diamond; *Figure 3B* right black bar).

## Representational untangling of orientation is specific to V1

To show that the RU of orientation information is specific to V1 and does not occur at earlier stages of visual processing, we performed simulations with two other model neuron populations in which selectivities of individual neurons resembled that of neurons in the retina and the lateral geniculate nucleus (LGN). For this we used neurons that were sensitive to a single pixel in the stimulus (as a simple model of photoreceptor activations in the retina; *Figure 3C*, inset) or neurons characterised by center-surround receptive fields (modelling retinal ganglion and LGN cells; *Figure 3D* inset). For a fair comparison with our V1 population, we used the same number of cells, with the same set of receptive field locations, and the same amount of overall signal and noise variability in their membrane potentials.

While linear decodability was similarly high as from V1 responses without phase nuisance (*Figure 3C–D*, black bars; cf. *Figure 3B*, black bars), it was markedly different once phase nuisance was introduced (*Figure 3C–D*, blue bars; cf. *Figure 3B*, blue bars). Not only was orientation linearly undecodable from the membrane potentials of our model retinal and LGN populations (*Figure 3C–D*, MP blue bars) but, in contrast to the V1 population, it also remained undecodable from their firing rates, even with the best possible choice of the firing threshold (*Figure 3C–D*, FR blue bars; see Appendix 1 for an intuition).

## Decoding with multiple nuisance features

Simple cells in V1 show mixed selectivity to a number of stimulus features beside orientation and phase. Thus, we extended our analyses to include two more nuisance features, spatial frequency (or its inverse, spatial period) and contrast, that are among the two strongest modulators of V1 responses (*Hubel and Wiesel, 1968*) and, in addition, they are analogous to size and illumination, which are in turn among the most commonly considered nuisance features for high-level RU (*Brincat and Connor, 2004*; *Ito et al., 1995*; *Vogels and Biederman, 2002*). We tested linear decoding performance with all eight possible combinations of these nuisance features varying or being fixed. When varied, each feature was sampled from a probability density that was chosen to reflect the main characteristics of natural stimulus statistics (*Figure 4A*, Materials and methods).

When fixing a nuisance feature, we chose a value that was near the mean of the natural distribution (*Figure 4A*, ticks on the x-axis).

Overall, the pattern of results was similar to that obtained with variability in phase only (*Figure 4*, black and blue for no variability in nuisance features, and variability in phase only, respectively, repeated from *Figure 3A,B*): more nuisance variability decreased performance (*Figure 4C*), and the trade-off between total information and sparseness resulted in a peak in performance at intermediate threshold levels (*Figure 4B*, coloured lines). Interestingly, we found the most often studied nuisance parameter to cause the least amount of representational entanglement: variability in contrast affected linear decodability of membrane potentials the least (*Figure 4C*, red) because the response manifolds corresponding to changes in contrast were radial lines (as membrane potential responses simply scaled with contrast) which all intersected only at one point and thus created little additional ambiguity (not shown). Importantly, while there was a slight variation in the optimal firing threshold values that allowed maximal decoding performance (*Figure 4B*, coloured ticks), this variation was relatively small across the eight combinations of nuisance features we tested and remained largely unchanged when the receptive field parameters of the neurons were perturbed (*Figure 4—figure supplement 1*).

To test the robustness of these results to variations in stimulus statistics, we also varied the distributions of spatial frequency and contrast (*Figure 5A–B*). (We left the uniform phase distribution unchanged as it is unlikely that any realistic stimulus manipulation would lead to particular phases to be overrepresented.) By using the original parameter distributions (*Figure 4A*) to construct training stimuli for the decoder and different parameter distributions to test performance, this analysis provided a test of how well the optimal thresholds generalised across different stimulus distributions. We found that the firing thresholds that were optimal for the original stimulus distributions remained near optimal with these changed distributions: there was no discernible loss of performance compared to a decoder which was trained with the modified stimulus distributions (*Figure 5C–D*, bars vs. horizontal lines).

While it is orientation coding that is most often associated with simple cell activity in V1, all other stimulus parameters to which simple cells show selectivity can be regarded as targets for decoding. Thus, to further test the robustness of our findings, we also studied the linear decodability of the spatial period or contrast of stimuli, while letting all other (nuisance) stimulus parameters (including orientation) vary (see Materials and methods). We found that the linear decodability of both stimulus parameters had a similar firing threshold-dependence as that of orientation, with a distinct peak performance close to the optimal threshold for orientation decoding (*Figure 4—figure supplement 2*; cf. *Figure 4*, grey).

## Parameter-dependence of the optimal firing threshold

We noted that for most model settings we studied so far (identity and distribution of fixed vs. variable nuisance stimulus features) there was a clear optimum for the firing threshold which remained roughly constant across all conditions (*Figure 4*). In contrast, simple scaling arguments predicted (Materials and methods) that performance should fundamentally depend on specific combinations of single cell parameters (see *Figure 2B*). In particular, it should depend on how much of the noise variability ($\sigma$) versus signal variability (controlled by the depth of modulation, $u_{\mathrm{AC}}$) in membrane potentials is 'removed' (i.e. mapped to zero firing rate) by the thresholding of the FRNL. In turn, the optimal threshold ($u_{th}^{opt}$) should shift with the mean membrane potential ($u_{\mathrm{DC}}$), and scale when noise and signal variance are jointly scaled. We tested these predictions by varying either the depth of modulation (*Figure 6A*, $\times$ symbols) or the noise variance of membrane potentials (*Figure 6A*, + symbols) and confirmed that in all these cases the appropriately normalised optimal threshold, $\left(u_{th}^{opt} - u_{\mathrm{DC}}\right)/u_{\mathrm{AC}}$, showed the same, approximately linear, relationship with the noise-to-signal ratio, $\sigma/u_{\mathrm{AC}}$ (*Figure 6A*).

To further test the robustness of this result, we systematically varied a number of other model parameters: the number of neurons in the population ($N$, *Figure 6B*), the number of orientation categories to be decoded ($K$, *Figure 6C*), the magnitude of noise correlations in the population ($\rho$, *Figure 6D*), the super-threshold shape (exponent) of the FRNL of neurons ($\kappa$, *Figure 6E*). Wherever possible, we varied these parameters around experimentally found values. For example, average noise correlations measured in spike counts in awake V1 are typically reported to be less than 0.1

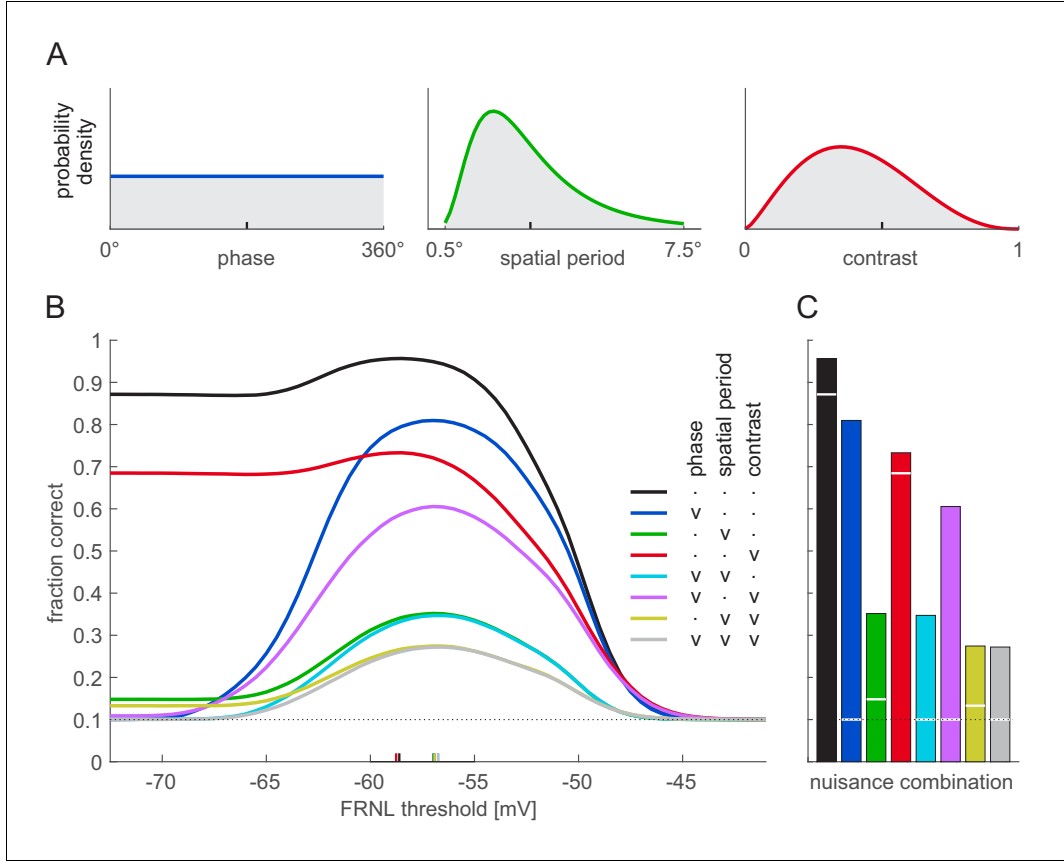

**Figure 4.** Effect of nuisance parameter variability on decoding performance. Nuisance parameters were varied individually (blue: phase; green: spatial period; red: contrast) or in combination (mixture colours). (**A**) Parameter distributions used for varying nuisance parameters: uniform for stimulus phase (left), lognormal for spatial period (middle), and beta distribution for contrast (right). Ticks on x-axes show parameter values used when the corresponding parameter was fixed. (**B**) Linear decoding performance as a function of the FRNL threshold for different combinations of variable nuisance parameters (colours, see legend for details: 'v' denotes variable, '.' denotes fixed parameter). Coloured ticks on x-axis show optimal thresholds for the corresponding combinations of variable nuisance parameters (i.e. the locations of peaks on the corresponding performance curves). Note that even when all nuisance parameters are variable, a linear decoder performs above chance around the optimal threshold. (**C**) Linear decoding performance for firing rates obtained with optimal FRNL threshold values under different nuisance parameter uncertainties (bars, colours as in B) and for membrane potentials (white lines). Note that the only nuisance parameter that shows membrane potential decoding performance considerably above chance is contrast (red). Simulations without nuisance parameter variability (black) and with only phase variability (blue) are replotted from *Figure 3A*.

DOI: https://doi.org/10.7554/eLife.43625.008

The following figure supplements are available for figure 4:

**Figure supplement 1.** Variance in decoding performance as a result of variations in receptive field parameters of model neurons.

DOI: https://doi.org/10.7554/eLife.43625.009

**Figure supplement 2.** Orientation, spatial period, and contrast decoding from firing rate responses.

DOI: https://doi.org/10.7554/eLife.43625.010

(*Ecker et al., 2011*) which imply average membrane potential correlations (which we are modelling here) in the range of 0.06– 0.15 (*Bányai et al., 2017*) (*Figure 6D*). The average exponent of the suprathreshold part of the FRNL of V1 simple cells was found to be ~1.2 (*Carandini, 2004*) (*Figure 6E*). For other parameters, those controlling the size of the population (*Figure 6B*), and the resolution of the estimation task (*Figure 6C*), experimentally validated data was not available and so we varied them several-fold to ensure our results remained robust to them. In all cases, we measured

linear decoding performance while varying phase as the nuisance stimulus feature (as in *Figure 4B*, blue).

We found that a clear optimum existed for the firing threshold in all cases (*Figure 6—figure supplement 1*). Although peak performance could depend strongly on some model parameters (*Figure 6—figure supplement 1*), the value of the optimal threshold at which it was achieved was largely independent of these parameters (*Figure 6*). Specifically, as expected, peak performance increased with the number of cells in the population (*Figure 6—figure supplement 1A*) and decreased with the number of orientation categories to be decoded (*Figure 6—figure supplement 1B*), but, the scaling of the optimal threshold with the noise-to-signal ratio remained invariant to either parameter (*Figure 6B–C*). Both peak performance (*Figure 6—figure supplement 1C*) and the optimal threshold (*Figure 6D*) were only weakly affected by increasing correlations among the responses of the neurons (either uniformly across the population, or such that they had a specific 'information limiting' structure; *Figure 6—figure supplement 2*, see also *Moreno-Bote et al. (2014)* and the Appendix 1). The only other parameter that had a substantial effect on the optimal threshold (but not on peak performance, *Figure 6—figure supplement 1D*) was the exponent of the FRNL (*Figure 6E*).

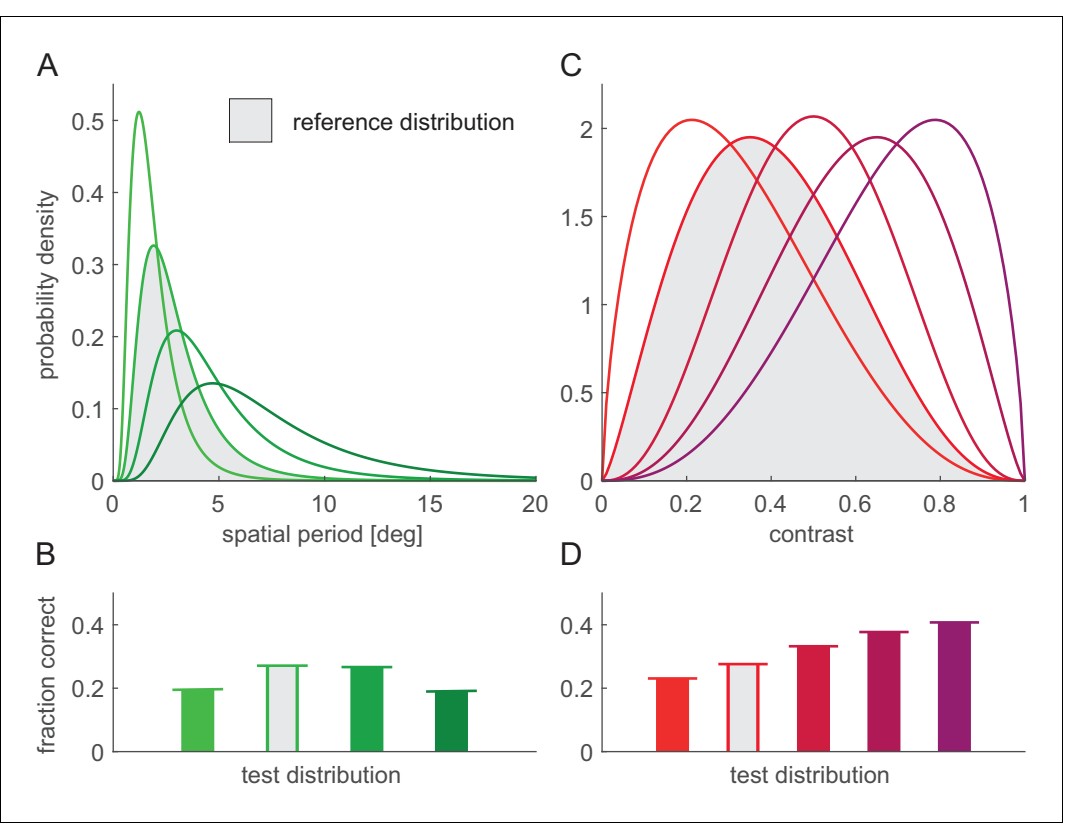

**Figure 5.** Robustness of the optimal FRNL threshold to changes in nuisance parameter statistics. (**A**) Distributions of spatial period. Grey-shaded distribution is the reference distribution used in other figures where spatial period was a nuisance parameter. (**B**) Performance of the linear decoder under different period distributions (colours as in A) with the 'default' FRNL threshold which was optimised to the reference spatial period distribution (bars). As an upper bound, performance with the FRNL threshold re-optimised for each spatial period distribution is also shown (horizontal lines). Note that bars reach horizontal lines in all cases, indicating that performance with the default FRNL threshold is indistinguishable from that achieved with the re-optimised FRNL threshold. (**C–D**) Same as (**A–B**) but for changes in the distribution of stimulus contrast.
DOI: https://doi.org/10.7554/eLife.43625.011

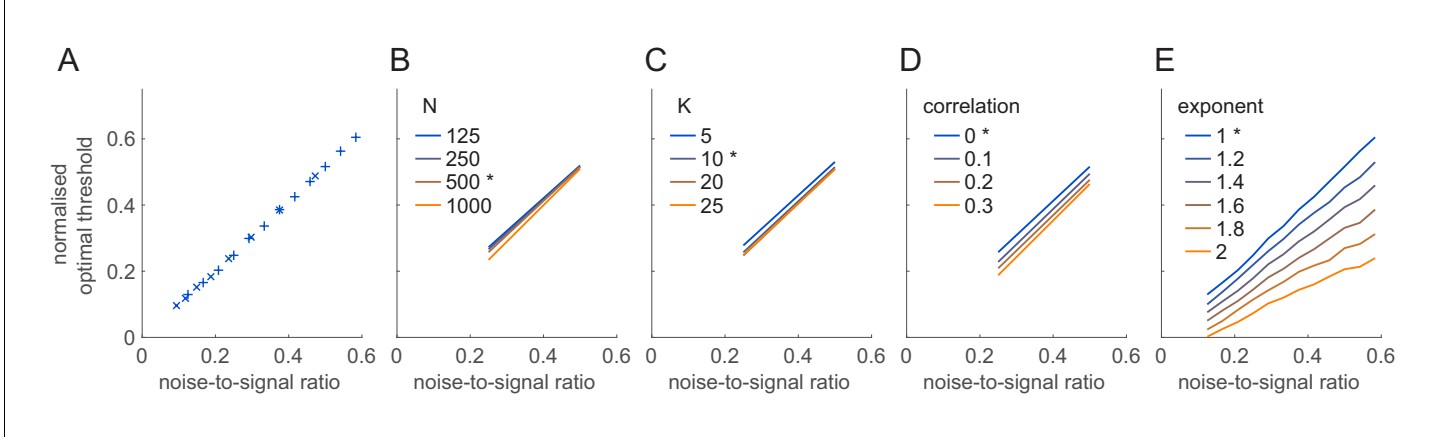

**Figure 6.** Dependence of the optimal threshold on model parameters. (A) Dependence of the normalised optimal threshold (y-axis) on the noise-to-signal ratio (x-axis). The normalised optimal threshold is the deviation of the optimal threshold from the baseline membrane potential, normalised by the magnitude of membrane potential noise: $(u_{th}^{opt} - u_{DC})/u_{AC}$ (see text for details). Noise-to-signal ratio is the ratio between the magnitude of noise and the strength of the signal: $\sigma/u_{AC}$. Note the same linear dependence irrespective of whether signal strength (×) or noise magnitude is varied (+). (B–E) Same as (A) for different population sizes (B), decoder resolutions (C), correlation strengths (D) and exponents of the FRNL nonlinearity (E). Asterisks mark the default values of the parameters used in other figures. Scaling of the optimal threshold with noise-to-signal ratio is largely independent from changes in these network and cellular parameters except for the exponent of the FRNL nonlinearity which substantially changes the noise-dependence of the optimal threshold.

DOI: https://doi.org/10.7554/eLife.43625.012

The following figure supplements are available for figure 6:

**Figure supplement 1.** Parameter-dependence of optimal FRNL threshold.
DOI: https://doi.org/10.7554/eLife.43625.013

**Figure supplement 2.** Effect of information-limiting correlations on decoding performance.
DOI: https://doi.org/10.7554/eLife.43625.014

## V1 simple cells have near optimal firing thresholds

The finding that the optimal firing threshold depended on only a handful, mostly directly measurable parameters of cellular responses allowed us to test experimentally whether the FRNL of simple cells supports RU in V1. For this, we studied the example of linear decoding of orientation with phase as a nuisance parameter. First, we estimated the mean, the modulation depth (both in mV) and the noise variance (in mV$^2$) of membrane potential responses, as well as the threshold of the FRNL (in mV) from intracellular recordings of V1 simple cells in awake mice viewing drifting full-field sinusoidal grating stimuli (i.e. with phase changing systematically; Materials and methods, *Figure 7A–B*, *Figure 7—figure supplement 1*). We then constructed model neuron populations with matching membrane potential response properties (using the experimentally measured mean, modulation depth and noise variance parameters) and for each model population computed the optimal threshold. Given our results on the parameter dependence of the optimal threshold (*Figure 6*), in order to compare experimentally measured and optimal threshold values, we expressed both in normalised units (subtracting the mean membrane potential and dividing by modulation depth) and plotted them as functions of the noise-to-signal ratio (the standard deviation of noise divided by modulation depth). As our data did not allow the reliable estimation of the precise value of the exponent of the FRNL (*Figure 7B*), we expressed our predictions for each value of the noise-to-signal ratio as the set of normalised threshold values that would result in at least 90% peak performance at any value of the exponent within the range of exponents (between 1 and 2) that earlier reports considered realistic (*Carandini, 2004*) (*Figure 7—figure supplements 2–3*). We found that the firing thresholds determined experimentally were in this near-optimal robust performance regime in all cases despite large differences in individual parameters across the cells we recorded (*Figure 7C*; blue circles). Randomly swapping parameters across cells revealed that the incidence of measured thresholds in the robust performance regime was significant (*Figure 7C* inset, black dots; permutation test, p=0.01),

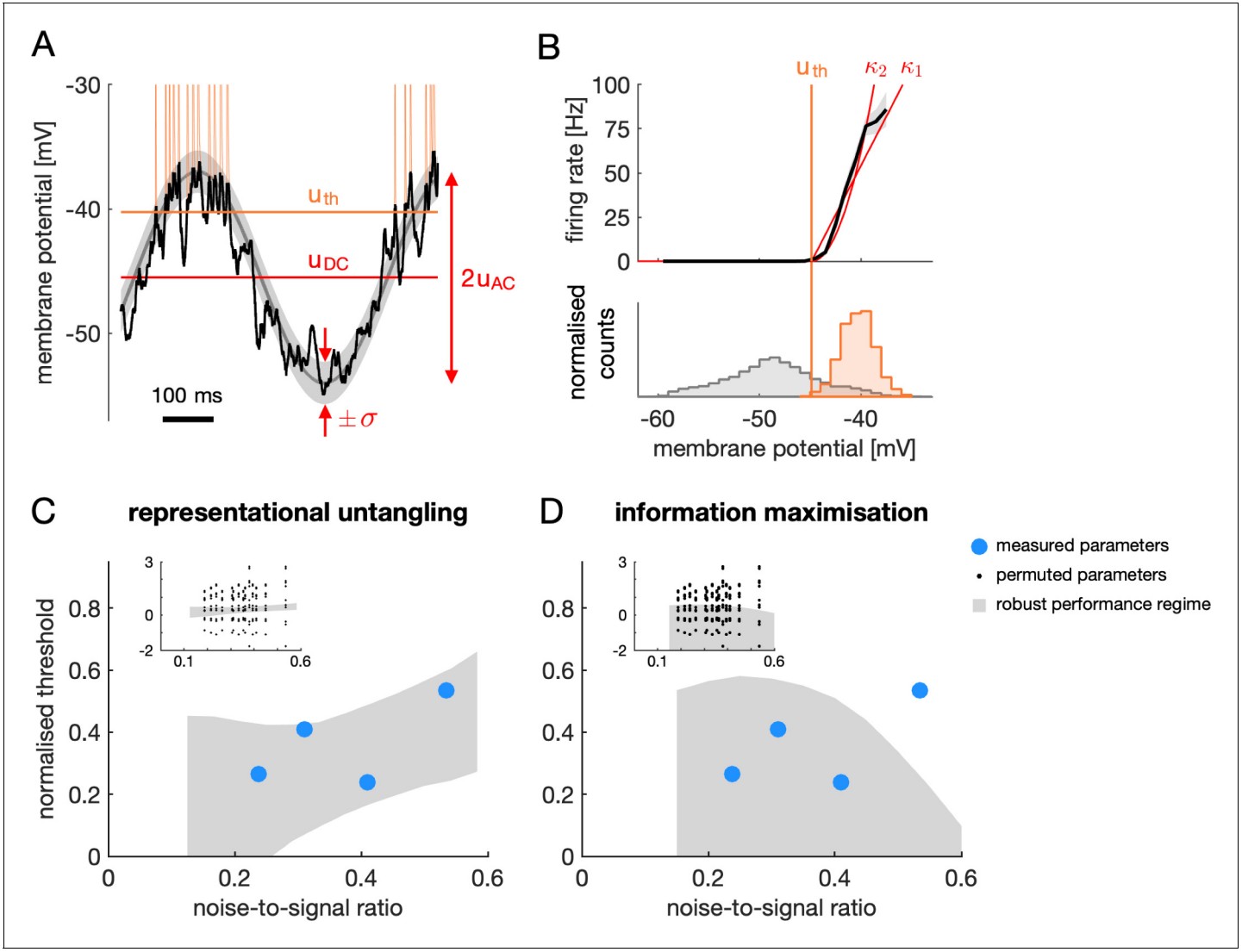

**Figure 7.** Comparison of optimal firing thresholds derived from the model and firing thresholds of V1 simple cells. (**A**) Intracellularly recorded 'generator' membrane potential trace of a V1 simple cell (black) with spikes clipped off (see Materials and methods). Orange vertical lines show spike times, orange horizontal line shows estimated firing threshold ($u_{th}$). Red marks show response baseline ($u_{DC}$), signal variance ($u_{AC}$), and noise magnitude ($\sigma$) estimated from the generator potential responses to multiple cycles of the moving grating stimulus. (**B**) Illustration of estimating the FRNL from electrophysiological data. The firing rate of a cell as a function of the membrane potential (top panel, black line, gray shaded area shows s.e.m.) is obtained as the normalised ratio of the probability distributions of the generator potential at spike times and at all times (bottom panel, orange and gray histograms, respectively). A threshold-power-law function was fitted to the firing rate function and the threshold linear fit was used to estimate the firing threshold (vertical orange line). Threshold-power-law fits with different levels of the FRNL exponent (red traces, $\kappa_1 = 1$, $\kappa_2 = 2$) were largely consistent with the firing rate estimated from the data. (**C**) Normalised thresholds of four V1 simple cells as a function of the noise-to-signal ratio of their membrane potential responses (blue circles). Normalised thresholds and noise-to-signal ratios were computed from the original cellular parameters (panels A-B) as in *Figure 6*. Shaded area shows robust decodability regime of RU (range of normalised thresholds within which > 90% of maximal linear decoding performance is achieved across all $\kappa$ values between 1 and 2, cf. *Figure 6E*, *Figure 7—figure supplement 2*). Inset: same as main panel but black dots correspond to normalised threshold values and noise-to-signal ratios obtained by shuffling the experimentally measured parameters across recorded neurons ($u_{DC}$ = −46.8, −42.4, −47.3, −36.9 mV; $u_{AC}$ = 7.15, 9.1, 4.48, 7.83 mV; $\sigma$ = 1.7, 2.83, 2.39, 3.21; and $u_{th}$ = −44.9, −38.6, −44.9, −35.1 mV). (**D**) Same as panel (**C**) but recorded data is compared with the prediction of maximal information transmission on the normalised threshold.

DOI: https://doi.org/10.7554/eLife.43625.015

The following figure supplements are available for figure 7:

**Figure supplement 1.** Spike fitting.

DOI: https://doi.org/10.7554/eLife.43625.016

**Figure supplement 2.** Dependence of the optimal threshold on the level of membrane potential noise.

*Figure 7 continued on next page*

*Figure 7 continued*

DOI: https://doi.org/10.7554/eLife.43625.017

**Figure supplement 3.** Robust decodability of population responses.

DOI: https://doi.org/10.7554/eLife.43625.018

suggesting that the near-optimal thresholds we found in the actual cells required specific co-tuning of these parameters.

These analyses also offered a way to directly compare our hypothesis that V1 is optimised for RU to the classical infomax hypothesis that V1 is optimised for total information transmission. Recall that infomax predicts that the optimal firing threshold is as low as possible. In principle, this optimum is well below the average membrane potential, which appears to be in contradiction with our experimental data that showed firing thresholds clearly above the average membrane potential (*Figure 7C*, normalised thresholds are all above 0). However, we also found that total information remained at ceiling for substantially higher values of the firing thresholds (*Figure 3*, optimal decoder), raising the possibility that infomax may also be able to account for our data. Thus, we followed the same approach as for RU, and rather than concentrating on the unique optimal threshold, which was difficult to define without knowing the precise value of the FRNL exponent, we identified the robust performance regime as the set of normalised threshold values that would result in at least 90% maximal total information at each value of the noise-to-signal ratio, and across the whole regime of realistic FRNL exponents (*Figure 7D*, grey region). Only 3 out of 4 of our recorded cells were in this regime (*Figure 7D*, blue circles), and given the breadth of the robust performance regime extending to infinitely low thresholds, this was not significant by using the same permutation test as in the case of RU (p=0.54). Thus, in contrast to RU, the specificity of the infomax hypothesis was too limited to be able to convincingly account for the data.

## The effect of population heterogeneity

The results above were obtained assuming a homogeneous population of neurons which only differed in their receptive field locations, preferred orientations and phases, but had otherwise identical parameters. As population heterogeneity can have an important influence on neural coding (*Ecker et al., 2011*; *Shamir and Sompolinsky, 2006*), we also studied heterogeneous populations. In particular, we were wondering whether our finding of near-optimal thresholds in individual cells (*Figure 7*) would be representative even if those cells were part of a heterogeneous population. Therefore, we constructed a population in which noise variance ($\sigma^2$) and FRNL exponent ($\kappa$) were varied across cells in experimentally found ranges (*Carandini, 2004*). The FRNL threshold of each individual cell was then set to a value that would have been optimal in a homogeneous population in which all cells had the same parameters ($\sigma^2$ and $\kappa$) as that single cell. (That is, we only optimised thresholds 'locally' for each neuron, rather than attempting to find the globally optimal combination of thresholds for the population of cells). We then randomly varied the thresholds of all cells around their respective locally optimal values and asked whether there were parameter combinations which yielded better performance. We found that modifying the thresholds generally resulted in deteriorating performance, such that decoding performance was highest near the original locally optimal thresholds (*Figure 8*). Thus, the thresholds found to be optimal based on the assumption of a homogeneous population still provided high performance in more realistic, heterogeneous populations. This result also suggests that overall ('global') optimality of population decoding performance might be achieved by some optimization rule acting locally on the firing threshold of each neuron separately, without needing information about the cellular parameters of other neurons, which would be difficult to obtain by biologically plausible mechanisms.

We also studied the effect of additional forms of heterogeneity in individual receptive field properties (size, non-circularity or inclination, preferred spatial frequency) by adjusting these parameters to approximately match those found in experiments (*Jones and Palmer, 1987*) and found that this also had only a small influence on performance, or the optimal threshold (data not shown).

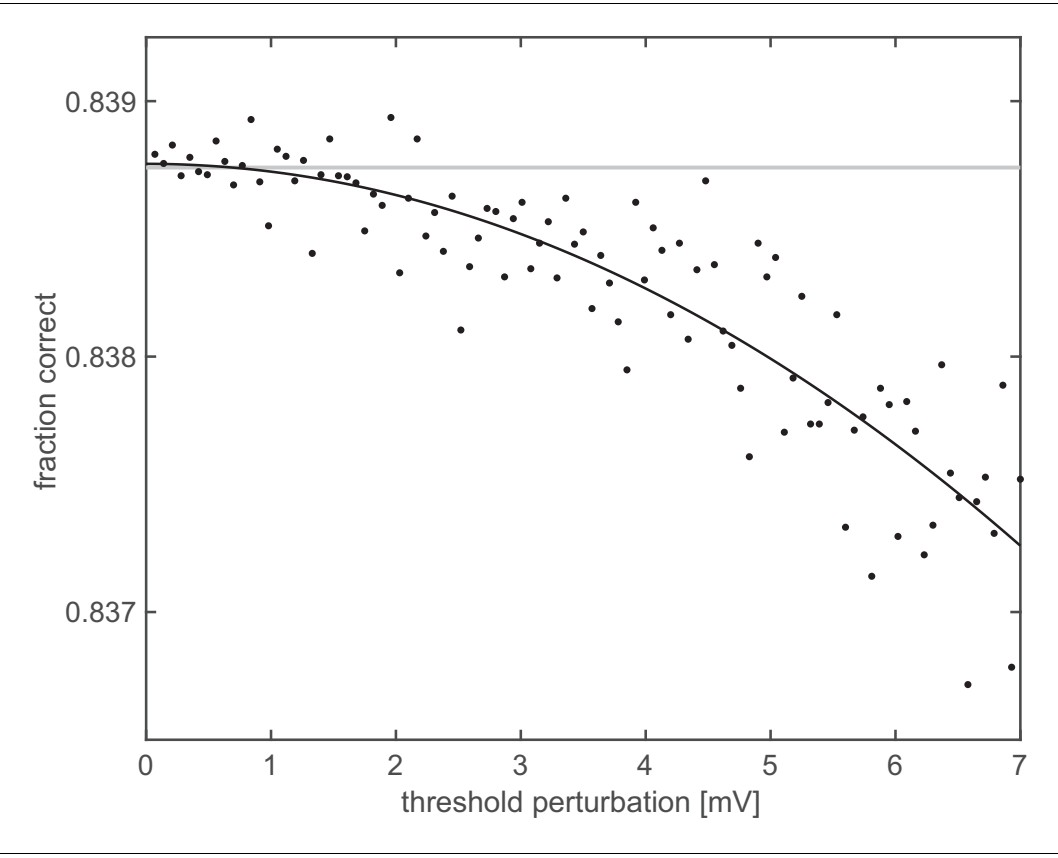

**Figure 8.** Representational untangling in a heterogeneous population. Orientation decoding performance from a population (N = 500) of neurons with heterogeneous properties as a function of the magnitude of deviation (Euclidean distance) from the optimal firing thresholds determined based on the assumption of a homogeneous population. Both the exponent of the FRNL and the level of membrane potential noise were chosen from a set of five possible values (1, 1.2, 1.4, 1.6, 1.8, and 2, 2.5, 3, 3.5, 4 mV, respectively), yielding 25 different parameter combinations. Thus, twenty neurons with different receptive fields were assigned to each parameter combination. For each of the 25 parameter combinations, the optimal firing threshold was established by simulating a homogeneous population with those parameters. In the heterogeneous population, the threshold for each cell was set to the value optimised for the corresponding homogeneous population. Decoding performance was measured after perturbing these thresholds (black dots). Grey line shows decoding performance with the unperturbed thresholds, black line is a quadratic fit to the perturbed performance levels.

DOI: https://doi.org/10.7554/eLife.43625.019

## Discussion

We have shown that the firing rate nonlinearity (FRNL) of V1 simple cells contributes to the representational untangling (RU) of orientation and other low-level visual information. While decoding performance is traditionally considered to be limited by the 'noise' variability in neural responses (*Berens et al., 2012*; *Chen et al., 2006*), our focus on RU warranted that we explicitly took into account another, oft-neglected source of response variability: that due to variability in nuisance parameters of the stimulus, such as phase, contrast, and spatial period. We have quantified RU by the linear decodability of membrane potentials or firing rates, and found that despite the obvious (and substantial) information loss entailed by the FRNL, sparsification of responses made the format of information in firing rates more amenable to linear decoding. Our analyses suggested this effect to be specific to V1 as it did not arise in model populations of retinal or LGN cells with non-oriented receptive fields. We also found that the value of the FRNL threshold that struck the optimal balance between sparsifying responses and preserving information was robust to variations in the identity of decoded and nuisance features, and in fact most other model parameters, and depended only on a few, experimentally well-defined local (as opposed to population-wide) quantities characterising the

responses of individual cells. An analysis of intracellular recordings of mouse V1 simple cells showed that the thresholds of these cells were near optimal for RU despite substantial variability in their cellular parameters. In comparison, an alternative computational objective that is often considered to be relevant for V1, information maximisation, was unable to specifically account for these data. These results suggest that the FRNL of V1 simple cells may be specifically adapted to support the RU of orientation information.

## Visual processing in ecologically relevant regimes

Although the evolutionary objective of the visual system is to maximise performance on natural images, we used highly simplified, full-field sinusoidal gratings as stimuli (but note that natural image statistics were taken into account in the choice of the distribution of nuisance parameters; *Figure 5A*). Our choice for artificial stimuli was motivated by a number of factors. First, it allowed our results to be directly compared to a large swathe of the theoretical and experimental literature that used the same stimuli (*Ecker et al., 2011*; *Seung and Sompolinsky, 1993*; *Shamir and Sompolinsky, 2006*; *Berens et al., 2012*; *Gutnisky et al., 2017*). Second, it also allowed us to show that nuisance parameters, rather than the traditionally studied factors of single-neuron variability or noise correlations, are the main bottleneck for decoding (and that this bottleneck is at least partially alleviated by the firing rate nonlinearity) even for the same simple stimuli that previous studies have used, without considering the whole complexity of natural images. Third, more complex stimuli will recruit mechanisms based on lateral and feed-back connections (eg. those responsible for extra-classical receptive field effects) that the network architecture we used here cannot capture. However, our main results remained essentially unchanged when we considered a 'hypercolumnar' population representing *local* rather than *global* orientation (ie. such that all cells had their receptive fields in the same location; *Figure 3—figure supplement 3*). Importantly, inasmuch as our model represents an appropriate approximation of at least such a hypercolumnar population, the content of natural images outside this (classical) receptive field location will not affect the decoding of the content at this location, and thus these results should generalise to natural stimuli.

In general, the performance of any (but the optimal) decoder depends on the amount of data used to train it. Indeed, we often need to make decisions based on only a few training examples – something that biological learning systems excel at (*Lake et al., 2015*). However, in contrast to high-level cognitive tasks requiring flexible decision making, it is reasonable to expect that the decoding of low-level visual features in an early visual area, such as V1, has been optimised on evolutionary time scales and would thus not be limited by the amount of data experienced over the lifetime of an individual. Therefore, in all cases, we trained our linear decoders with sufficient amounts of data so that to achieve asymptotic performance. This meant that we mostly tested generalisation performance only for new instances of membrane potential noise, not for new stimuli (but see *Figure 5* for generalisation to new distributions of stimuli.) Nevertheless, this approach also allowed us to demonstrate that even in the limit of infinite training data, nuisance parameters represent a fundamental challenge for RU that can be mitigated by the appropriate firing rate nonlinearity.

## Decoding from firing rates versus spike counts

In previous work, decoding was often performed from spike counts rather than firing rates (*Berens et al., 2012*; *Pitkow and Meister, 2012*; *Seung and Sompolinsky, 1993*; *Shamir, 2014*; but see *Abbott and Dayan, 1999*; *Ecker et al., 2011*; *Shamir and Sompolinsky, 2006*). The transformation between the two entails further information loss due to the discrete nature of spike counts and potential additional (Poisson) stochasticity in them, with the magnitude of this information loss depending on the time window used for counting spikes. However, in the limit of large time windows or large populations, variability in firing rates due to nuisance parameter variability dominates over the effects of spiking variability. While the relevant time window for decoding may depend on the ecological situation and the specific task an animal is facing, the size of the population is likely large enough to allow the effective averaging out of spiking variability. Importantly, we have also shown that population size only scales overall performance but does not affect the value for the optimal threshold (*Figure 6—figure supplement 1*). Moreover, the Gaussian variability in membrane potentials with deterministic conversion to firing rates we assumed, combined with a deterministic spike generation process, has been shown to result in spike count variability that is

phenomenologically similar to classical Poisson spike count models (*Carandini, 2004*) and in fact matches experimentally observed stimulus (orientation and contrast) dependent changes in spike count (co-)variability better (*Bányai et al., 2017*). Thus, we expect our results to generalise to spike count decoding from large experimentally recorded populations.

## Noise correlations

We found an increase, albeit relatively small, in decoding performance with an increase in noise correlations (*Figure 6D*, showing results for uniform noise correlations – similar results, not shown, were obtained with other correlation structures). Although this may at first seem counterintuitive (correlations imply redundancy), it is well known that the effects of noise correlations depend on their relation to the tuning of cells (*Averbeck et al., 2006*; *Lin et al., 2015*) and they generally increase linear Fisher information for the particular (tuning-independent) pattern of correlations we studied (*Abbott and Dayan, 1999*). As expected, information limiting correlations decreased the performance of both the optimal Bayesian decoder and of the linear decoder, such that the efficiency of the linear decoder relative to the optimal decoder became higher – that is the resulting code was relatively more linearly decodable. While other patterns of correlations may result in a decrease of performance, noise correlations in V1 tend to be small overall (*Ecker et al., 2011*). Moreover, as we argued above, *effective* noise correlations will be dominated by correlations induced by nuisance parameter variability in the settings we studied, and so the effects of 'standard' noise correlations are likely to be diminishing.

## Complex cells in V1

Orientation selectivity is a central feature of V1 neurons (*Hubel and Wiesel, 1968*). We argued that the mixed selectivity of neurons affects the linear decodability of stimulus information adversely: if neurons are selective to additional stimulus features then variations in these will likely cause representational entanglement. Stimulus phase is particularly prone to causing representational entanglement of orientation information if neuronal responses are jointly modulated by both phase and orientation. Importantly, the level of phase selectivity greatly varies across neurons (*Niell and Stryker, 2008*; *Skottun et al., 1991*). For a neuron that is sensitive to orientation but not to phase, a so-called complex cell, variability in stimulus phase is not detrimental and decodability remains intact. It is unclear if the FRNL-based mechanism of RU contributes to the emergence of such complex cells, but there is a suggestive correspondence between the canonical model of complex cells and the architecture we studied. According to the canonical model of complex cell responses, these responses are brought about by a specific pooling of simple cell responses. Intriguingly, the mathematical form of this pooling is essentially isomorphic to the linear decoder we studied here: it takes a linear combination of the non-linearly (typically quadratically) transformed responses of a number of simple cells differing in their preferred phases (*Hubel and Wiesel, 1968*) and potentially other receptive field properties (*Rust et al., 2005*). This suggests that the same principles that we found determine the optimal firing threshold of simple cells for an abstract linear decoder may also determine the optimal firing threshold of simple cells for the efficient operation of complex cells. Furthermore, our simulations show similar detrimental effects for nuisance parameters other than phase, including spatial period. For these other nuisance parameters, complex cell properties have limited capacity to prevent the detrimental effect of entanglement. We argue that the FRNL provides a surprisingly effective solution for the more general problem of decoding under nuisance parameter uncertainty.

## The computational role of the FRNL

We have shown that the FRNL has an important computational role in RU. Previous work implicated the FRNL of V1 cells in achieving contrast-invariant orientation tuning curves in V1 simple cells (*Finn et al., 2007*). This effect can also be understood as a spacial case of a mechanism promoting RU: contrast-invariant tuning curves contribute to more efficient contrast-invariant decoding of firing rates by ensuring that firing rates are simply scaled by contrast and so decision boundaries for orientation decoding (*Figure 2*) remain radial and thus linear when contrast varies (*Ma et al., 2006*). However, contrast is but one of several nuisance parameters whose variability makes RU of orientation information challenging in V1, and as we have shown, other nuisance parameters (phase and spatial

period) have even more dramatic effects (*Figure 4*). Our results thus extend previous work by placing contrast invariant tuning curves in the wider context of RU and showing that the FRNL of simple cells plays a general role in keeping orientation information linearly decodable in the face of variability in a number of nuisance parameters.

The FRNL has been shown to contribute to performing linear classification on arbitrary mappings of a set of variables (*Barak et al., 2013*). In such tasks, no single input feature alone can be used for solving the task by linear read-out. Thus, mixed selectivity (i.e. the property that neuronal responses depend on multiple stimulus attributes, *Asaad et al., 1998*; *Churchland and Shenoy, 2007*; *Warden and Miller, 2010*) was shown to be advantageous and even necessary (e.g. in higher-order association cortices during tasks requiring cognitive flexibility, *Rigotti et al., 2013*). In contrast, in the standard task of orientation decoding in V1 considered here, an 'orderly' input-output mapping is required, in which one input feature needs to be mapped to the output monotonically. For this task, the mixed selectivity of neurons is less of a blessing: if neurons had pure selectivities for orientation such that their responses did not depend on any other stimulus parameters then variation in nuisance parameters would not lead to representational entanglement. Moreover, earlier work on the effects of response sparsification on linear decodability studied a network of abstract binary neurons (*Barak et al., 2013*) and thus could not relate sparseness to a biophysically well-defined firing threshold as we did. Our results generalise the utility of the FRNL, showing that it pertains even to an elementary sensory decoding task.

Given that the main direct effect of the FRNL is the sparsification of neural responses (*Figure 3A*, red), one might then intuitively reason that setting the FRNL threshold to very high values to achieve ultra-sparse codes could increase linearly separability even more. In this limit, each image would be coded as a one-hot population response vector. Although one-hot population coding achieved by ultra-sparse codes has appealing theoretical properties, it fundamentally relies on assuming no noise, and requires an exponential number of neurons (in the number of nuisance parameters). In fact, in the (unrealistic) limit of no noise, the question of an optimal threshold even becomes somewhat moot as essentially all thresholds above a minimum will perform equally well (essentially perfectly) – as the broadening of the robust performance regime towards low values of the noise-to-signal ratio in *Figure 7C* (and *Figure 7—figure supplement 3*) also suggests. Moreover, we expect ultra sparse coding to be particularly sensitive to contrast as a nuisance parameter (as the correct threshold for achieving a one-hot code will critically depend on the overall scaling of responses which in turn depends monotonically on contrast, such that selecting a single optimal threshold is impossible). Importantly, the conditions for ultra-sparse codes are also unlikely to be met in real V1, for example the experimentally measured levels of noise-to-signal ratio in our V1 data were well above 0 (*Figure 7*). Indeed, at these realistic noise levels, we found that increasing the number of neurons in the population did not favour higher thresholds which could have potentially led to such ultra sparse codes (*Figure 6B*).

The FRNL-induced increase in sparseness has also been shown to contribute to increasing mutual information between visual stimuli and the responses of retinal ganglion cells (*Pitkow and Meister, 2012*), such that there was an optimum for the FRNL threshold at intermediate values. Although our results may superficially suggest a similar interpretation, they are in fact orthogonal. It is important to note that, in our case, the performance of the optimal decoder (the analogue of mutual information measured by *Pitkow and Meister, 2012*) was a monotonically decreasing function of the FRNL threshold, without an optimum at intermediate values. The difference is due to the fact that *Pitkow and Meister (2012)* modeled the effects of the FRNL threshold in a regime in which spiking noise dominated. Specifically, they studied small populations of neurons ($N \leq 8$) and kept the average firing rate of neurons constant (by adjusting their peak firing rate) as the FRNL threshold was varied. This meant that a decrease in the threshold in their setting led to sustained firing with low spike counts, which were associated with high relative variability, thus diminishing information in the low threshold regime. In contrast, we considered a large population of neurons in which spiking noise is less relevant (and thus decoding performance does not depend on the overall scaling of firing rates) and instead the effects of nuisance parameters dominate (see above). Indeed, in line with our results, *Pitkow and Meister (2012)* also found that increasing population size shifted the value of the FRNL threshold at which total (mutual) information was maximised towards smaller values, such that the dominant effect was now a decrease in total information for higher thresholds. Thus, the optimal intermediate value for the FRNL threshold we found emerged for a fundamentally

different reason: because we measured linearly decodable information rather than total information, which is brought about by a trade-off between total information and sparseness. Taken together, these results suggest that the FRNL threshold may play an important role in neural computations at different stages of sensory processing via different mechanisms: by maximising total information transmission in the retina, and by achieving RU in the visual cortex.

## Materials and methods

### Population model of MP responses

The default population model for encoding stimuli consisted of $N = 500$ simple cells whose membrane potential responses were established by calculating circular Gabor filter responses plus Gaussian noise (*Figure 1B*). Each circular Gabor filter (indexed by n) is described by six parameters: the coordinates of the center of the filter measured from the line of sight $(x_n, y_n)$, the spatial period of the plane wave component of the Gabor filter ($\lambda$), the orientation of the sinusoidal component ($\theta_n \in [0, 180°)$), the phase offset of the sinusoidal component relative to the center of the filter ($\varphi_n \in [0, 360°)$), standard deviation of the circular Gaussian envelope ($\delta$). All angles are measured in degrees, retinal distances (coordinates, spatial period, envelope width) are measured in degrees of visual angle. Numerical values of the above filter parameters were chosen according to *Table 1*. In the default model, spatial period and envelope width were identical across the population and identical parameters were defined to be approximately equal to the empirical average from *Jones and Palmer (1987)*. Filter locations were uniformly sampled in the whole visual field (90 degrees), which ensured that not only local stimulus effects were considered. Preferred orientations and phases of Gabor filter were uniformly sampled from the entire range of possible values (*Figure 3—figure supplement 1A*). Code used for simulation is available on GitHub (*Gáspár, 2019*; copy archived at https://github.com/elifesciences-publications/representational_untangling).

In order to limit simulation time and eliminate discretisation noise, analytical filter responses are calculated in response to sine wave stimuli. A strictly finite size realistic receptive field requires truncation of the receptive field but such a truncation would prevent the analytical calculation of filter responses, therefore only the standard exponentially decaying Gaussian envelope of the Gabor-filter is supposed to keep the filter localised. Filter response of a circular Gabor filter with infinite domain to an infinite sine wave stimulus when spatial period of the filter and the stimulus are supposed to be identical is given by:

$$
\begin{aligned}
&u_{\text{DC}} + u_{\text{AC}} \cos(\psi + \varphi - \phi) \exp\left[-\left(\tfrac{2\pi\delta}{\lambda}\right)^2 (1 - \cos(\theta - \vartheta))\right] + \\
&u_{\text{AC}} \cos(\psi + \varphi - \phi) \exp\left[-\left(\tfrac{2\pi\delta}{\lambda}\right)^2 (1 + \cos(\theta - \vartheta))\right]
\end{aligned}
\tag{1}
$$

Here response is shifted and scaled such that the predefined value $u_{\text{DC}}$ matches the phase averaged DC component and $u_{\text{AC}}$ matches the amplitude of the phase modulated AC component at the preferred orientation at 100% contrast level (for derivations in the case of unequal spatial periods see the Appendix 1). In the above expression $\lambda$ is the common spatial periods; $\vartheta$: grating stimulus orientation; $\phi$: stimulus phase relative to the line of sight;

$$
\psi = 2\pi[\sin(\vartheta)x - \cos(\vartheta)y]/\lambda
\tag{2}
$$

a phase term belonging to the filter location. A similar expression can be obtained for more general settings of a grating stimulus (not shown). Deviation of the above analytical response from the response of a truncated Gabor filter (to a local stimulus) is not substantial. Subscript indices of filter parameters, identifying a particular filter of the population are not shown for clarity.

Parameters of the Gabor filters determine the mean response characteristics of a model neuron: the mean membrane potential was assumed to be equal to the linear filter response to a stimulus. As a consequence, mean responses of individual neurons to oriented grating stimuli can be characterised by a tuning curve with peak response corresponding to the preferred orientation of the neuron. The response of a neuron was the sum of the mean response and a Gaussian noise. The noise, however, was not necessarily independent: in some experiments (*Figure 6D*) correlation between

membrane potential responses was introduced by using multivariate Gaussian noise to determine the responses of the complete population. Noise was assumed to have no temporal structure beyond a limited time window, therefore samples were considered to be iid samples in 20 ms time bins. Amplitude and contrast parameters of stimuli are kept constant throughout the simulations, filter responses are scaled and shifted to match empirical neural responses, here characterised by $u_{\mathrm{DC}}$ and $u_{\mathrm{AC}}$ (*Table 1*). Noise level was set to be 25% relative to the signal variance (matched to *Carandini, 2004* example simple cell).

## Population models for LGN and retina responses

To study the potential contribution of FRNL to RU at earlier stages of the visual processing hierarchy, we defined LGN and retina models by altering the filter properties of the encoding population. Retina receptive fields were approximated by pixel responses, while LGN receptive fields were approximated by difference of Gaussian filters (DoG, *Dayan and Abbott, 2005*). In order to be able to use analytic calculations derived for Gabor filters, we approximated DoG responses as the difference of two constrained Gabor filters. The sinusoidal component of the Gabor filter was modified such that the phase relative to the Gabor center ($\varphi_n$) was zero and the spatial period was increased so that within the Gaussian envelope there was no practical modulation. The sizes of the central ON-region and peripheral OFF-region were set to 1.5° and 4°, respectively. To be able to contrast retina, LGN and V1 analyses, filter positions were matched to those of the V1 population and retina/LGN filter responses were calibrated such that stimulus variance of the model neurons were equal to their corresponding neurons in V1.

## Mapping of membrane potentials to firing rates

The nonlinear transformation from membrane potential to firing rate is described by the firing rate nonlinearity (FRNL). A general threshold-power-law function, $r(u) = \Phi[u - u_{th}]_{+}^{\kappa}$, is used to simulate simple cell firing responses that has empirically been found to fit simple cell responses well (*Carandini, 2004*). In this expression $[.]_{+}$ indicates rectification. The specific value of the scaling factor $\Phi$ does not affect results. The value of the power-law exponent for the simulations was $\kappa = 1$ (except *Figure 6* and *Figure 6—figure supplement 1*) and the range of possible physiological values (*Carandini, 2004*) was explored when fitting membrane potential recordings (*Figure 7*). Parameters of the FRNL are assumed to be identical across cells (but see *Figure 8*). Since nonlinearity is central to our analysis FRNL threshold was varied in the simulations in order to study its effect on decoding performance.

## Linear decoder

A single-layer decoder was used to perform probabilistic linear decoding of orientation information from the stimulus. The decoder represented $K$ classes and was performing multinomial logistic regression by assigning probabilities to the represented classes. Weights, which could take both positive and negative values, were tuned to be optimal by supervised learning on a set of static training stimuli (*Figure 3—figure supplement 1B,C*). In the training data set multiple values of the decoded parameter belonged to any given class ($M_{\vartheta}$ in the case of orientation). Training and testing was performed on different stimulus sets, and training and testing samples differed in membrane potential noise. Wherever nuisance parameters were present, parameters of training images were sampled from the distribution characteristic of the nuisance parameter(s). Parameters of individual test images were sampled from the same distribution, except for *Figure 5* where a different parameter distribution was used for one of the nuisance parameters. Stimulus parameter distributions were constructed such that these approximated the characteristics of natural images. Fitting of the weight parameters of the decoder was performed in MATLAB using a custom code that uses the Barzalai-Borwein method to perform gradient descent (*Barzilai and Borwein, 1988*; *Gáspár, 2019*). Training stimulus parameter space is uniformly covered by a 2D grid in case of phase uncertainty, and the same grid is used to test the decoder. $M_{rep}$ (and $M_{rep'}$) describes the number of repetitions with newly generated noise to generate the whole training (or testing) stimulus data set (*Table 1*). For any given nuisance parameter that is characterised by non-uniform prior (see *Figure 4*) a distorted multidimensional grid is used to generate the stimulus bank. For spatial period, a lognormal distribution with parameters $\mu = 0.95$, $\sigma = 0.55$ was used. For contrast, a beta-distribution was fitted to local contrast

**Table 1.** Parameters of the decoding model under phase uncertainty and their default values are shown or their generator methods are indicated (first column).

Description of the parameters and methods are expounded (second column). The performance curve of the above described standard model is shown by the thick blue curve on *Figure 3A* and *Figure 4B* and used as a reference simulation on *Figures 5* and *6* where parameters of the model are varied.

**Parameters of the encoder Gabor population**

| | |
|---|---|
| $N = 500$ | Number of Gabor cells in the encoder population (with the exception of *Figure 6B*) |
| $x_n, y_n \sim$ uniform over a disk<br>$R = 90°$ | coordinates of the center of Gabor filter n measured from the line of sight; randomly chosen from a uniform distribution over a disk with radius R (with the exception of *Figure 3—figure supplement 3*) |
| $\lambda = 3°$ | period of the plane wave component of Gabor filters; identical for all cells (with the exception of *Figure 4—figure supplement 2*) |
| $\theta_n[°] = 180° \cdot (n-1)/N \in [0°, 180°)$ | preferred orientation of cell n; evenly distributed on the entire range |
| $\varphi_n[°] \in [0°, 360°)$ | phase offset of the sinusoidal wave component of the Gabor filter relative to the center; evenly distributed on the entire range, but the order is randomly permuted to avoid correlation with the preferred orientation (with the exception of no-nuisance simulations where $\varphi_0 = 180$) |
| $\delta = 2°$ | standard deviation of the circular Gaussian envelope of Gabor filters; identical for all cells |

**Parameters of the rescaling of filter responses to membrane potential values**

| | |
|---|---|
| $u_{DC} = -60 \, [\text{mV}]$ | mean value of phase modulated filter responses |
| $u_{AC} = 12 \, [\text{mV}]$ | peak amplitude of maximally modulated filter responses; numeric value is chosen as a typical value (*Carandini, 2004*) (but varied on *Figure 6A*) |

**Variability and covariability of membrane potential responses**

| | |
|---|---|
| $\sigma = \xi \cdot u_{AC} = 3 \, [\text{mV}], \xi = 25\%$ | std of Gaussian membrane potential noise is measured relative to the signal amplitude; identical for all cells (but varied on *Figures 6*, *7C* and *8*) |
| $\Sigma_{ij} = \rho_{ij} \cdot \sigma^2 = 0, i \neq j$ | off-diagonal elements of the $N \times N$ covariance matrix; no correlation structure is assumed (with the exception of *Figure 6*) |

**Firing rate nonlinearity**

| | |
|---|---|
| $\kappa = 1$ | power-law exponent of the FRNL (but varied on *Figures 6E*, *7C* and *8*) |
| $u_{th} \, [\text{mV}]$ | threshold of the FRNL; this is always a running variable in the simulations |
| $\Phi = 16.7 \, [\text{Hz}]$ | prefactor of the FRNL of Gabor cells |

**Parameters of the categorisation task**

| | |
|---|---|
| $K = 10$ | number of discrete orientation categories (varied only on *Figure 6C*) |

**Parameters of the stimulus set used for training the decoder**

| | |
|---|---|
| $M_\vartheta = 10$ | number of bins for stimulus orientation, $\vartheta$, within an orientation category (represents orientation uncertainty) |
| $M_\phi = 50$ | number of bins for stimulus phase, $\phi$ (represents phase uncertainty) |
| $M_{rep} = 20$ | number of stimulus repetitions at a given $(\phi, \vartheta)$ with independent noise |
| $M = K \times M_\vartheta \times M_\phi \times M_{rep} = 10^5$ | total number of stimuli used for training the decoder |
| $\lambda_s = 3°$ | Spatial period of sine wave stimuli; matched to the spatial period of the Gabor filters (but see *Figures 4* and *5* and respective captions) |
| $c = 0.5$ | contrast of the sine wave stimuli; chosen to be the mean value of the natural distribution used later in *Figure 4* or *Figure 6* (but see *Figure 5*) |
| $\vartheta_m[deg] \in [0, 180)$ | orientation of the grating stimulus m $(m = 1, \ldots, M)$, 0 being the horizontal direction |
| $\phi_m[deg] \in [0, 360)$ | phase of grating stimulus m relative to line of sight; generated such that $(\vartheta_m, \phi_m)$ pairs come from the $M_\vartheta \times M_\phi$ rectangular grid covering uniformly the $\vartheta$–$\phi$ parameter space |

**Parameters of the stimulus set used for testing the decoder**

| | |
|---|---|
| $M_{rep'} = 50$ | number of stimulus repetitions with given $(\vartheta, \phi)$; other parameters of the testing stimulus bank is the same as the training parameters above |

DOI: https://doi.org/10.7554/eLife.43625.020

distribution based on a small dataset containing 24-hour time-lapse images, resulting in parameters $\alpha = 2.4$, $\beta = 3.6$.

The resulting decoder provided class probabilities, and the maximum class probability was used to indicate the decision. We tested the robustness of our claims by using alternative methods for

decision. These analyses revealed variations in the measured level of decoding performance but the optimal threshold was invariant to the choice of performance measure (*Figure 3—figure supplement 2A*). The efficiency of the decoding is measured by fraction correct.

## Optimal decoder

The optimal decoder was constructed by inferring class labels from data by explicitly inverting the process of the generation of output from the population of neurons. Membrane potential responses of individual neurons for a grating stimulus with a particular orientation and phase was described by normal distribution

$$p(u_n|\vartheta,\varphi) = N\left(u_n; g_n(\vartheta,\varphi),\sigma^2\right), \tag{3}$$

where $g_n(\vartheta,\varphi)$ is the Gabor-filter response and $\sigma^2$ is the level of membrane potential noise. Likelihood of observing a particular firing rate response is

$$p(r_n|\vartheta,\varphi) = p_n^0(\vartheta,\varphi)\delta(r_n) + H(r_n)\rho_n(r_n|\vartheta,\varphi), \tag{4}$$

where $p_n^0(\vartheta,\varphi)$ is the baseline probability of zero firing rate response, $H()$ is the Heaviside function and $\rho_n(r_n|\vartheta,\varphi) = p(u_n|\vartheta,\varphi)(dr/du)^{-1}$. Assuming a discrete set of stimulus phases, the posterior probability of orientation $\vartheta_k$ is

$$p(\vartheta_k \mid \underline{r}) = \sum_{j=1}^{J}\prod_n p\left(r_n|\vartheta_k,\varphi_j\right)p(\vartheta_k)p(\varphi_j)/p(\underline{r}). \tag{5}$$

Priors, $p(\vartheta)$ and $p(\varphi)$, were chosen to be uniform.

An optimal decoder was derived when ILC was assumed to contribute to the membrane potential noise. We constructed the decoder by explicitly modelling how ILC is introduced: ILC can be modelled by assuming that the value of the decoded variable (the orientation in our case) is not constant but changes stochastically. We assumed that the orientation of stimuli, $\vartheta$, are sampled from a normal distribution around the true value, $\Theta$:

$$p(\vartheta \mid \Theta) = N\left(\vartheta;\Theta,\sigma_\vartheta^2\right). \tag{6}$$

In order to obtain the posterior probability for the true orientation, a marginalisation over the stochastically sampled orientations beyond the marginalisation over phases is required:

$$p(\Theta \mid \underline{r}) = \sum_\vartheta p(\Theta,\vartheta \mid \underline{r}) = \sum_\vartheta p(\vartheta \mid \underline{r})p(\vartheta \mid \Theta). \tag{7}$$

## Contrast and spatial period decoding

Orientation decoding is a standard measure of stimulus encoding in V1 simple cells but the joint selectivity to a number of other variables, including contrast and spatial period means that the contribution of FRNL to decoding these variables can also be directly tested. Diversity of the filter properties of the encoding population is crucial for efficient decoding. When studying orientation decoding properties of the population we ensured diversity in filter orientation and phases but used identical spatial period sensitivities across the population. When studying spatial period and contrast decoding, we constructed a population which represented the spatial period characteristics of the stimuli by matching the distribution of period of the filters by sampling the spatial periods from the period distribution of the stimuli. Such an encoding population was used in all of the analyses in *Figure 4—figure supplement 2*. Here, after choosing a particular parameter for decoding, all other stimulus parameters were regarded as nuisance parameters. Classes for the decoded stimuli were established by partitioning the decoded parameters according to the cumulative distribution of the stimuli: stimulus classes were defined as the centers of the partitions containing equal probability mass.

## Infomax model

The objective of the infomax model is the optimisation of information transmission with respect to the decoding variable, which is formulated through the mutual information:

$$I(\vartheta, \underline{r}) = \sum_{k=1}^{K} \int d\underline{r}\, p(\vartheta_k, \underline{r}) \log \frac{p(\vartheta_k, \underline{r})}{p(\vartheta_k)p(r)} =$$
$$\sum_{k=1}^{K} \int d\underline{r}\, p(\underline{r} \mid \vartheta_k)\, p(\vartheta_k) \log p(\vartheta_k \mid \underline{r}) - \sum_{k=1}^{K} p(\vartheta_k) \log p(\vartheta_k) \tag{8}$$

where the first term on the right hand side is the conditional entropy of the class label given possible responses and the second term is the entropy of the category distribution, the index $k$ runs through the different orientation classes. Note that the conditional probability under the logarithm in the first term is the posterior of the class label. The second term in *Equation 8* does not depend on the response distribution therefore it is not affected by changes in firing properties of the neurons, including the firing threshold.

The probabilistic fraction correct (PFC) performance measure,

$$\mathrm{PFC} = \left( \prod_{m=1}^{M} p(\vartheta_m | \underline{r}) \right)^{1/M}, \tag{9}$$

where $m$ runs through the trials of the experiment, is in an intimate relationship with the mutual information (*Equation 8*). This can be seen by taking the logarithm of PFC is

$$\log PFC = \frac{1}{M} \sum_{m=1}^{M} \log p(\vartheta_m | \underline{r}) \tag{10}$$

which corresponds to a Monte Carlo approximation to the integral in the conditional entropy in *Equation 8* since in individual trials stimuli and population responses to the particular stimulus can be regarded as samples:

$$\vartheta_m \sim p(\vartheta_m)$$
$$\underline{r} \sim p(\underline{r} | \vartheta_m). \tag{11}$$

This indicates that the PFC performance measure is an approximation to exponent of the mutual information, up to a scaling constant that is determined by the entropy of the stimuli (*Figure 3—figure supplement 2B*).

## Experimental subjects and surgical procedures

All experimental procedures were approved by the University of California Los Angeles Office for Protection of Research Subjects and the Chancellor's Animal Research Committee. 1–12 month old C57Bl6/J mice underwent implantation of head-bars, surgery recovery, acclimation to the spherical treadmill and craniotomy over V1 as described in *Polack et al. (2013)*. A 3 mm diameter coverslip drilled with a 500 μm diameter hole was placed over the dura such that the coverslip fit entirely in the craniotomy and was flush with the skull surface.. The coverslip was maintained in place using Vetbond and dental cement and the recording chamber was filled with cortex buffer containing 135 mM NaCl, 5 mM KCl, 5 mM HEPES, 1.8 mM CaCl2 and 1 mM MgCl2. The head-bar was fixed to a post and the mouse was placed on the spherical treadmill to recover from anesthesia. All recordings were performed at least 2 hr after the end of anesthesia.

## Electrophysiological recordings

Two-photon guided in-vivo whole-cell recordings were performed as described in *Polack et al. (2013)*. Long-tapered micropipettes made of borosilicate glass (1.5 mm outer diameter, 0.86 mm inner diameter, Sutter Instrument) were pulled on Sutter Instruments P-97 pipette puller to a resistance of 3–7 MΩ, and filled with an internal solution containing 115 mM potassium gluconate, 20 mM KCl, 10 mM HEPES, 10 mM phosphocreatine, 14 mM ATP-Mg, 0.3 mM GTP, and 0.01–0.05 mM Alexa-594. Whole-cell current-clamp recordings were performed using the bridge mode of an Axoclamp 2A amplifier (Molecular Devices), further amplified and low-pass filtered at 5 kHz using a Warner Instruments amplifier (LPF 202A). Series of current pulses of small intensity (typically −100 pA) were used to balance the bridge and compensate the pipette capacitance. The membrane potential was not corrected for liquid junction potentials (estimated to be about 10 mV).

## Visual presentation

Visual stimuli were presented as described in *Polack et al. (2013)*. A 40 cm diagonal LCD monitor was placed in the monocular visual field of the mouse at a distance of 30 cm, contralateral to the craniotomy. Custom-made software developed with Psychtoolbox in MATLAB was used to display drifting sine wave gratings (single orientations at the preferred orientation, temporal frequency = 2 Hz, spatial frequency = 0.04 cycle per degree, contrast = 100%). The presentation of each orientation lasted 3 s, which was preceded by the presentation of a gray isoluminant screen for an additional 3 s.

## Data processing of intracellular measurements

MP changes of V1 simple cells induced by drifting grating stimulus were fitted with three parameters: MP DC level ($u_{DC}$), signal variance ($u_{AC}$), level of noise ($\sigma$). Further, FRNL threshold was also extracted from MP recordings for each neuron individually. First, a spike removal algorithm is applied on the raw MP data to obtain the generator potential (*Carandini, 2004*) and the time of action potentials. Our spike removal algorithm used composite analytical fits (*Figure 7—figure supplement 1*) to accurately subtract spikes from raw MP data. This allowed precise estimation of residual MP levels at the location of spikes, which was required for FRNL calculation. Since the phase of periodic membrane potential modulation shows some variance across trials, and there are trials in which the phasic modulation of the membrane potential is difficult to determine, we performed the analysis in such a way that individual cycles of the periodic modulation are extracted from the 3-sec duration of stimulus presentation (6 cycles). The three parameters of the membrane potential response were calculated individually for these trials from the corresponding data segments. To do so, multiple cycles from a given trial were overlaid. The DC level (in mV) was established by averaging across cycles and time; AC level (in mV) was established by averaging across the overlaid cycles and measuring the amplitude of the average modulation; noise level (in mV$^2$) was measured by calculating variance across overlaid cycles.

For the estimation of the parameters of FRNL, the method described by *Carandini (2004)* was used (*Figure 7B*). The trial-averaged generator potential was segmented using 20 msec windows, and mean membrane potential levels in these bins were used to construct a histogram with 1 mV bin size. Another histogram was constructed for the number of spikes generated at any given membrane potential level. To obtain the mean firing rate corresponding to a membrane potential level, spike counts were normalised with the total duration of membrane potential segments in that bin. A threshold-linear function was fitted to the above determined FRNL to estimate the FRNL threshold.

## Acknowledgements

This work was supported by a Lendület Award of the Hungarian Academy of Sciences (GO), an award from the National Brain Research Program of Hungary (NAP-B, KTIA_NAP_12-2-201, GO), and the Wellcome Trust (ML), the Whitehall Foundation (PG) and NIH R01 grant MH105427 (PG) and the Human Frontier Science Program (RGP0044/2018, PG,ML,GO,). We would like to thank Michael Einstein for assistance with electrophysiological recordings. The authors declare no competing financial interests.

## Additional information

### Funding

| Funder | Grant reference number | Author |
|---|---|---|
| Hungarian Academy of Sciences | Lendulet Fellowship | Merse E Gáspár<br>Gergo Orban |
| Wellcome Trust | | Merse E Gáspár<br>Máté Lengyel |
| Human Frontier Science Program | RGP0044/2018 | Peyman Golshani<br>Máté Lengyel<br>Gergo Orban |

| National Institutes of Health | R01 MH105427 | Peyman Golshani |
|---|---|---|
| Whitehall Foundation | | Peyman Golshani |
| National Brain Research Program of Hungary | 2017-1.2.1-NKP-2017-00002 | Gergo Orban |

The funders had no role in study design, data collection and interpretation, or the decision to submit the work for publication.

## Author contributions
Merse E Gáspár, Conceptualization, Software, Formal analysis, Validation, Investigation, Visualization, Methodology, Writing—original draft, Writing—review and editing; Pierre-Olivier Polack, Conceptualization, Resources, Data curation, Software, Validation, Investigation, Methodology; Peyman Golshani, Resources, Data curation, Supervision, Funding acquisition, Methodology, Project administration; Máté Lengyel, Gergő Orbán, Conceptualization, Formal analysis, Supervision, Funding acquisition, Visualization, Methodology, Writing—original draft, Project administration, Writing—review and editing

## Author ORCIDs
Pierre-Olivier Polack (iD) http://orcid.org/0000-0003-1716-6595
Máté Lengyel (iD) http://orcid.org/0000-0001-7266-0049
Gergő Orbán (iD) https://orcid.org/0000-0002-2406-5912

## Ethics
Animal experimentation: All animal experiments were approved by University of California Los Angeles IACUC and Animal Research Committee (Protocol #06-066).

## Decision letter and Author response
Decision letter https://doi.org/10.7554/eLife.43625.024
Author response https://doi.org/10.7554/eLife.43625.025

# Additional files

## Supplementary files
• Transparent reporting form
DOI: https://doi.org/10.7554/eLife.43625.021

## Data availability
Data is available along with code at the GitHub repository https://github.com/CSNLWigner/representational_untangling (copy archived at https://github.com/elifesciences-publications/representational_untangling).

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

**Appendix 1**

DOI: https://doi.org/10.7554/eLife.43625.022

## 1 Intuition for the specificity of representational untangling of orientation decoding to V1

The reason for the difference between V1 and these upstream areas in the ability of the FRNL to help linear decodability can be understood through a simple intuition. The main effect of the FRNL is distinguishing between stimuli based on whether the corresponding neural responses exceed the firing threshold or not. This is useful for orientation decoding with phase nuisance when cells can attain different maximal membrane potential values depending on stimulus orientation even when pooling across all possible stimulus phases. For Gabor filter receptive fields this is clearly the case: at the preferred orientation of a cell, it will be deeply modulated by phase, and will attain a high maximal membrane potential value at its preferred phase, while at the orthogonal orientation it will be unmodulated by stimulus phase and will thus only attain intermediate values of the membrane potential. If the firing threshold is between these intermediate and maximal values, the FRNL can contribute to decoding. In contrast, cells with non-oriented receptive fields, such as those we used to model upstream areas, will show the same amount of membrane potential modulation at any stimulus orientation, and thus the attained maximal membrane potentials will not differ between stimulus orientations. As a result, no firing threshold will be able to distinguish between different orientations, and linear decoding performance remains at chance, just as we found.

## 2 Information-limiting correlations

Imposing a flat correlation structure on the membrane potential responses had a small effect on peak performance (*Figure 6—figure supplement 1C*) and on optimal threshold (*Figure 6D*). The effect of correlations, however, depends not only on magnitude but also on its specific structure: Information Limiting Correlation (ILC) (*Moreno-Bote et al., 2014*) imply covariations on neuron populations responses that are indistinguishable from those caused by changes in the decoded stimulus, stimulus orientation in our case and are known to have detrimental effects on coding. We investigated the effect of ILC by introducing orientation noise in the stimulus: the orientation of input stimulus was randomly sampled from a five degree mormal distribution. The 'private' noise added to the linear filter responses (*Figure 1*) was rescaled so that the total variance, the joint effect caused by the variance in stimulus drive and that of the 'private' noise, remained the same as in earlier simulations (3 mV). ILC limits decoding performance by introducing uncertainty about the identity of the stimulus orientation, which can be seen in decreased Bayesian decoder performance too (*Figure 6—figure supplement 2*). Importantly, while ILC also affected linear decoding by reducing overall performance, the qualitative properties of the firing rate decoder remained intact: the decoder was characterised by a clear optimal threshold and the location of the optimum was similar to the independent noise case. Interestingly, the magnitude of decline in linear decoding performance relative to the performance of the optimal decoder is smaller when variance is partly caused by ILC than in the case of a network where noise is independent across neurons (90% vs. 78%, respectively).

## 3 Analytical calculation of Gabor filter responses to sine grating stimuli

Gabor filter response ($G(x, y)$) to general sine wave stimulus,

$$S(x, y) = S_0 + S_1 \sin(px + qy + r) \tag{S1}$$

is calculated in a coordinate system aligned to the two cardinal axes of the Gabor filter. In this coordinate system the general form of a Gabor filter is given by

$$G(x,y) = G_0 \exp(-ax^2 - by^2)\cos(ux + vy + w)$$

(S2)

and the response of the filter is

$$I = \int_{-\infty}^{\infty} \int_{-\infty}^{\infty} G(x,y)\, S(x,y)\, \mathrm{d}x\, \mathrm{d}y$$

(S3)

The integral is broken down into two terms according to the two terms of **Equation S1**

$$I_0 = G_0 S_0 \int_{-\infty}^{\infty} \int_{-\infty}^{\infty} \mathrm{e}^{-ax^2} \mathrm{e}^{-by^2} \cos(ux + vy + w)\, \mathrm{d}x\, \mathrm{d}y$$

$$I_1 = G_0 S_1 \int_{-\infty}^{\infty} \int_{-\infty}^{\infty} \mathrm{e}^{-ax^2} \mathrm{e}^{-by^2} \cos(ux + vy + w)\sin(px + qy + r)\, \mathrm{d}x\, \mathrm{d}y$$

We introduce $\tilde{I}_0(x,y)$ and $\tilde{I}_1(x,y)$ such that $I_0 = G_0 S_0 \tilde{I}_0$ and $I_1 = G_0 S_1 \tilde{I}_0$, where we omitted the variables $x$ and $y$ for the clarity of the notation. We focus on the calculation of $\tilde{I}_1$ since $\tilde{I}_0$ can be obtained by substituting of $p = q = 0$ and $r = \pi/2$. Expansion of trigonometric functions results in sixteen terms, only four of which yield non-zero double integrals:

$$
\begin{aligned}
\tilde{I}_1 = \ & +\cos(w)\sin(r) \int_{-\infty}^{\infty} \mathrm{d}x\, \mathrm{e}^{-ax^2}\cos(ux)\cos(px) \int_{-\infty}^{\infty} \mathrm{d}y\, \mathrm{e}^{-by^2}\cos(vy)\cos(qy) + \\
& +\cos(w)\sin(r) \int_{-\infty}^{\infty} \mathrm{d}x\, \mathrm{e}^{-ax^2}\sin(ux)\sin(px) \int_{-\infty}^{\infty} \mathrm{d}y\, \mathrm{e}^{-by^2}\sin(vy)\sin(qy) + \\
& -\sin(w)\cos(r) \int_{-\infty}^{\infty} \mathrm{d}x\, \mathrm{e}^{-ax^2}\cos(ux)\cos(px) \int_{-\infty}^{\infty} \mathrm{d}y\, \mathrm{e}^{-by^2}\sin(vy)\sin(qy) + \\
& -\sin(w)\cos(r) \int_{-\infty}^{\infty} \mathrm{d}x\, \mathrm{e}^{-ax^2}\sin(ux)\sin(px) \int_{-\infty}^{\infty} \mathrm{d}y\, \mathrm{e}^{-by^2}\cos(vy)\cos(qy)
\end{aligned}
$$

Using the addition and subtraction formulae for trigonometric functions,

$$\tilde{I}_1 = -\frac{\cos(w)\sin(r)}{4}\int_{-\infty}^{\infty} dx\, e^{-ax^2}\left[\cos((u-p)x)+\cos((u+p)x)\right]\cdot$$

$$\int_{-\infty}^{\infty} dy\, e^{-by^2}\left[\cos((v-q)y)+\cos((v+q)y)\right] +$$

$$+\frac{\cos(w)\sin(r)}{4}\int_{-\infty}^{\infty} dx\, e^{-ax^2}\left[\cos((u-p)x)-\cos((u+p)x)\right]\cdot$$

$$\int_{-\infty}^{\infty} dy\, e^{-by^2}\left[\cos((v-q)y)-\cos((v+q)y)\right] +$$

$$-\frac{\sin(w)\cos(r)}{4}\int_{-\infty}^{\infty} dx\, e^{-ax^2}\left[\cos((u-p)x)+\cos((u+p)x)\right]\cdot \tag{S4}$$

$$\int_{-\infty}^{\infty} dy\, e^{-by^2}\left[\cos((v-q)y)-\cos((v+q)y)\right] +$$

$$-\frac{\sin(w)\cos(r)}{4}\int_{-\infty}^{\infty} dx\, e^{-ax^2}\left[\cos((u-p)x)-\cos((u+p)x)\right]\cdot$$

$$\int_{-\infty}^{\infty} dy\, e^{-by^2}\left[\cos((v-q)y)+\cos((v+q)y)\right]$$

Using $\int_{-\infty}^{\infty} e^{-ax^2}\cos(mx)\,dx = \sqrt{\frac{\pi}{a}}e^{-\frac{m^2}{4a}}$, all the integrals can be be expressed in a closed form:

$$\tilde{I}_1 = +\frac{\pi}{4\sqrt{ab}}\cos(w)\sin(r)\left[e^{-\frac{(u-p)^2}{4a}}+e^{-\frac{(u+p)^2}{4a}}\right]\left[e^{-\frac{(v-q)^2}{4b}}+e^{-\frac{(v+q)^2}{4b}}\right] +$$

$$+\frac{\pi}{4\sqrt{ab}}\cos(w)\sin(r)\left[e^{-\frac{(u-p)^2}{4a}}-e^{-\frac{(u+p)^2}{4a}}\right]\left[e^{-\frac{(v-q)^2}{4b}}-e^{-\frac{(v+q)^2}{4b}}\right] +$$

$$-\frac{\pi}{4\sqrt{ab}}\sin(w)\cos(r)\left[e^{-\frac{(u-p)^2}{4a}}+e^{-\frac{(u+p)^2}{4a}}\right]\left[e^{-\frac{(v-q)^2}{4b}}-e^{-\frac{(v+q)^2}{4b}}\right] + \tag{S5}$$

$$-\frac{\pi}{4\sqrt{ab}}\sin(w)\cos(r)\left[e^{-\frac{(u-p)^2}{4a}}-e^{-\frac{(u+p)^2}{4a}}\right]\left[e^{-\frac{(v-q)^2}{4b}}+e^{-\frac{(v+q)^2}{4b}}\right]$$

After expansion and simplification the above equation results in

$$\tilde{I}_1 = \frac{\pi}{2\sqrt{ab}}\left\{\left[\sin(r)\cos(w)-\sin(w)\cos(r)\right]e^{-\frac{(u-p)^2}{4a}-\frac{(v-q)^2}{4b}} + \right.$$

$$\left. +\left[\sin(r)\cos(w)+\sin(w)\cos(r)\right]e^{-\frac{(u+p)^2}{4a}-\frac{(v+q)^2}{4b}}\right\}.$$

Finally this can be rewritten in a compact form

$$\tilde{I}_1 = \frac{\pi}{2\sqrt{ab}}\left[\sin(r-w)e^{-\frac{(u-p)^2}{4a}-\frac{(v-q)^2}{4b}}+\sin(r+w)e^{-\frac{(u+p)^2}{4a}-\frac{(v+q)^2}{4b}}\right].$$

Substituting $p = q = 0$ and $r = \pi/2$, we can obtain a closed form expression for $\tilde{I}_0$:

$$\tilde{I}_0 = \frac{\pi\cos(w)}{\sqrt{ab}}e^{-\frac{u^2}{4a}-\frac{v^2}{4b}}.$$

Up to this point, for mathematical convenience, sine and cosine waves were parametrised with the wave vectors $(u, v)$ and $(p, q)$ together with the phases $w$ and $r$. In general, wave vectors $(k_x|k_y)$ can be mapped onto the more commonly used spatial period ($\lambda$) and orientation ($\vartheta$) parameters through

$$k_x = \frac{2\pi}{\lambda}\sin(\vartheta), \ k_y = -\frac{2\pi}{\lambda}\cos(\vartheta).$$

In the above derivation we assumed the coordinate system to be centred on the centre of the Gabor filter. Since in a population of Gabor filters this simplifying assumption cannot hold for most of the filters, we relax the assumptions to have arbitrary centre for the Gabor filter. Under these conditions, a Gabor filter is characterised by eight parameters: $x_0$, $y_0$, $\vartheta_G$, $\sigma$, $\epsilon$, $\delta$, $\lambda_G$, $\varphi_G$, where $(x_0, y_0)$ is the centre of the Gaussian envelope, $\vartheta_G \in [0, \pi)$ is the orientation of its major axis measured from the horizontal axis of the reference coordinate system, $\sigma$ is the largest variance of the Gaussian in the direction of the major axes, $\epsilon$ defines the ratio of the minor and major axes, $\delta \in [0, \pi)$ is the relative orientation of the cosine wave component measured from the major axes, $\lambda_G$ is the period of the cosine, and $\varphi_G$ is its phase at the centre of the Gaussian. This set of parameters can be mapped to our original set of six parameters $(G_0, a, b, u, v, w)$ by

$$
\begin{aligned}
G_0 &= \frac{1}{2\pi\epsilon\sigma^2}, \ a = \frac{1}{2\sigma^2}, \ b = \frac{1}{2\epsilon^2\sigma^2}, \\
u &= \frac{2\pi}{\lambda_G}\sin(\delta), \ v = -\frac{2\pi}{\lambda_G}\cos(\delta), \ w = \varphi_G
\end{aligned}
$$

Analogously, $\tilde{I}_1$ can be also expressed in this coordinate system, using the parameters spatial period ($\lambda_S$), orientation ($\vartheta_S$), phase ($\varphi_S$), and amplitude ($S_1$). The spatial period and the amplitude are independent of the coordinate system but orientation ($\vartheta'$) and phase ($\varphi'$) undergo a transformation upon change in the coordinate system:

$$\vartheta' = \vartheta_S - \vartheta_G \tag{S6}$$

$$r = \varphi' = \frac{2\pi}{\lambda_S}[\sin(\vartheta_S)x_0 - \cos(\vartheta_S)y_0] + \varphi_S \tag{S7}$$

Taken together, the Gabor filter response with parameters $x_0$, $y_0$, $\vartheta_G$, $\sigma$, $\epsilon$, $\delta$, $\lambda_G$, $\varphi_G$, to a sinusoidal grating stimulus, parametrised by $S_0$, $S_1$, $\lambda_S$, $\vartheta_S$, $\varphi_S$, is

$$
\begin{aligned}
I_1 = \frac{S_1}{2}\Bigg[ &\sin(\varphi_0 + \varphi_S - \varphi_G)\,e^{-2\pi^2\sigma^2\left(\frac{\sin(\delta)}{\lambda_G} - \frac{\sin(\vartheta_S - \vartheta_G)}{\lambda_S}\right)^2 - 2\pi^2\epsilon^2\sigma^2\left(\frac{\cos(\delta)}{\lambda_G} - \frac{\cos(\vartheta_S - \vartheta_G)}{\lambda_S}\right)^2} + \\
&+ \sin(\varphi_0 + \varphi_S + \varphi_G)\,e^{-2\pi^2\sigma^2\left(\frac{\sin(\delta)}{\lambda_G} + \frac{\sin(\vartheta_S - \vartheta_G)}{\lambda_S}\right)^2 - 2\pi^2\epsilon^2\sigma^2\left(\frac{\cos(\delta)}{\lambda_G} + \frac{\cos(\vartheta_S - \vartheta_G)}{\lambda_S}\right)^2}\Bigg]
\end{aligned}
$$

$$I_0 = S_0\cos(\varphi_G)\exp\left[-2\left(\frac{\pi\sigma}{\lambda_G}\right)^2(\sin^2\delta + \epsilon^2\cos^2\delta)\right]$$

where $\varphi_0 = 2\pi[\sin(\vartheta_S)x_0 - \cos(\vartheta_S)y_0]/\lambda_S$.

