## [Decision Letter]

Thank you for submitting your article "Representational untangling by the firing rate nonlinearity in V1 simple cells" for consideration by *eLife*. Your article has been reviewed by two peer reviewers, one of whom is a member of our Board of Reviewing Editors, and the evaluation has been overseen by Joshua Gold as the Senior Editor. The following individual involved in review of your submission has agreed to reveal their identity: Jim DiCarlo (Reviewer #2).

The reviewers have discussed the reviews with one another and the Reviewing Editor has drafted this decision to help you prepare a revised submission.

Summary:

In this study, the authors examine the role of threshold nonlinearities (a model of the mechanisms that intervene between membrane potential and firing rate) in neurons in cortical area V1. They use a combination of simulations and a bit of experimental data to argue that those thresholds appear to be set to maximize the (population) linear decodability of stimulus orientation in the face of nuisance parameters (phase, contrast, and spatial frequency). This hypothesis is referred to as representational untangling (RU). While untangling is an essential component of the visual form processing pathway, the mechanisms that support it are incompletely described, and in that context, this paper addresses an important and interesting set of issues. The reviewers have identified a number of issues that need to be addressed but they believe that they can be addressed via further analyses and some re-writing.

Essential revisions:

1) What is/are the main alternative hypotheses (to RU) and which figures/results reject those alternatives?

The main setup of this study is that the "designer" of V1 has followed a particular optimization goal (= a particular RU variant). (I mean "designer" here in the general sense of evolution, development, etc.) I kept wishing that the authors would more directly pose and test at least one serious alternative optimization goal. The main one they seem to entertain is preservation of total information:

"These results suggest that information re-formatting, rather than maximisation, may already be a relevant computational goal for the early visual system."

But that alternative is dismissed as almost trivial (and some versions are), which leaves one wondering why a study is even needed. As the authors point out, thresholds can only remove information. The more interesting alternative optimizations have to do with preserving information in the face of energy costs or other forms of compression (e.g. sparseness).

Here is a first attempt at stating an alternative:

"However, despite the obvious loss of total information caused by the rectifying aspect of the FRNL, which is due to all membrane potential values below the firing threshold being mapped to zero firing rate…" But this is not at all obvious at the population level.

And then "While this drastic removal of a large fraction of responses can clearly lead to severe total information loss (Barak et al., 2013)," The key word here is "can." But does it?

For example, the total information from the population of retinal ganglion cells might be spread out across an overcomplete set of V1 neurons, such that, even with thresholds, the V1 population maintains the total information (i.e. no loss). Of course, even if that were true (which I don't believe), then one would still have to suggest what the reason for doing that overcomplete spread out might be (e.g. preservation of information about natural images (not total information), energy costs, etc.).

To be more concrete and constructive: The most interesting testable experimental question here is how the thresholds are set (i.e. under what possible optimization goal.) The main setup seems to go like this: the population of V1 encoders is fixed, the noise in the membrane potentials of those encoders is irreducible, and the parameter under (optimization) control of this system is the threshold. Under these assumptions, the question posed is, what optimization goal is the system using to set the u_th_ parameter for each neuron? The authors claim that it is the particular RU goal they setup (Figure 7). It would be stronger if they could show that it is not (or less) consistent with some other optimization goal. In both cases, the simulations would assume the same fixed number of neuron and irreducible noise. This would make the paper much stronger. Alternatively, perhaps the authors could consider a range of RU optimization hypotheses (e.g. separability of orientation, separability of contrast, etc.)? Or, similarly, optimization for RU under limited training examples for the decoders? (see point 3 below).

Similarly, the authors focused solely on RU of the orientation of sinusoidal gratings from the responses of model simple cells. Optimality of the firing rate threshold is defined in the context of untangling information about the orientation of sinusoidal gratings, absent any reference to why posing the problem this way makes sense, given that these neurons presumably evolved to process natural images.

2) Try to strengthen the presented evidence for RU from the image input to this level of V1 (spiking outputs of V1 simple cells).

Previous experimental results advocating for RU along the ventral stream have made comparisons at two points of processing (e.g. V4 population spike rates and IT population spike rates). It would strengthen the claims if the authors would do something similar across the key stage of processing they are focused on here. E.g. compare with pixels, center-surround (e.g. difference of gaussians (DOG) to approx. RGC and LGN responses).

We understand that some decodes were attempted on the simulated intracellular potential population and compared with the spiking outputs. That is good, but this seems a bit like comparing apples and oranges.

3) Better define the RU hypothesis (or alternative variants of it). In particular, the operational definition of linear separability is not agreed upon in the field, and different definitions tend to test different things. We suggest would be to explore these different operational variants (below) and decide if the results are robust to that, or if they point in an even more interesting direction.

Most notably: It appears that the linear decoder test that was applied was only a test of generalization to new noise samples (subsection “Linear decoder”), rather than generalization to unseen values of the stimulus parameters. Is this correct? But then the manuscript reads: "RU was quantified by the performance of a linear decoder which decoded stimulus orientation from membrane potentials or firing rates in the face of noise and variability in other (nuisance) parameters of the stimulus: phase, contrast, and spatial frequency (Figure 1, blue)." This might imply tests of generalization to new stimulus values, but it is unclear. (Side point, the paper must be significantly clarified around this issue, regardless of what was done.)

Assuming that the training data completely cover all the test data (i.e. generalization only to new noise samples), then it is surprising to see that this linear decodability test would not be partially successful with decoders on the pixel (+noise) input representation. (Note that the two dimensional examples such as Figure 2 provide useful intuition, but tend to hide the possibilities of successful linear separability that is possible in high dimensional population spaces.) But the intuition might be wrong here because the chosen nuisance variables might strongly push against this possibility. Clarification around this issue would strengthen the paper. That is, one RU possibility is that NO separating hyperplane exists in the population state space. Another RU possibility is that multiple separating hyperplanes exists, but that they require a very large number of training examples to discover – thus, in practice, they appear not to exist. The difference in these two variants of RU depends on the amount of training data and the amount of required generalization.

To summarize: the improved linear decodabilty of the V1 population relative to a pixel or DOG representation should be shown directly and the dependence on training sample size should be explored in some way. The key result might be: for limited number of training examples, the properly thresholded V1 decoding performance is higher than the pixel and DOG representations, but that result is not true in the limit of infinite training data. To be clear, either of the possibilities above are interesting and publishable, but they are impossible to disentangle right now.

Related: an intuitive solution to increase linearly separability is to set RELU thresholds very high such that the representation of each image is ultra-sparse. In the limit, each image would be coded as one-hot population response vector. Under some operational definitions of linear separability, this solution is perfectly fine (e.g. test only at the training images). However, it is not a real world solution because it has little to no generalization power and it requires a very large number of neurons to maintain information and corresponding large numbers of training examples. Thus, the trick for real world tests of linear separability in the setting explored in this paper is to set the thresholds high enough to enable linear separability in the context of limited training data and test generalization to new nuisance values of the stimulus parameters – what that limit should be and how "far" the "new" stimuli should be are admittedly unclear, but both could be considered and explored to firm up the arguments. Perhaps this has already been done and it was missed it in the presentation.

---

## [Author Response]

Essential revisions:1) What is/are the main alternative hypotheses (to RU) and which figures/results reject those alternatives?[…] To be more concrete and constructive: The most interesting testable experimental question here is how the thresholds are set (i.e. under what possible optimization goal.) The main setup seems to go like this: the population of V1 encoders is fixed, the noise in the membrane potentials of those encoders is irreducible, and the parameter under (optimization) control of this system is the threshold. Under these assumptions, the question posed is, what optimization goal is the system using to set the u_th_ parameter for each neuron? The authors claim that it is the particular RU goal they setup (Figure 7). It would be stronger if they could show that it is not (or less) consistent with some other optimization goal. In both cases, the simulations would assume the same fixed number of neuron and irreducible noise. This would make the paper much stronger. Alternatively, perhaps the authors could consider a range of RU optimization hypotheses (e.g. separability of orientation, separability of contrast, etc.)? Or, similarly, optimization for RU under limited training examples for the decoders? (see point 3 below).

Thank you for the very constructive suggestion. Although, as the reviewer points out, we already referred to a specific alternative non-RU-related optimisation goal (information maximisation) in the previous version of the manuscript, and we thought it was clearly ruled out by the data, we did not unpack why we thought this was the case and, thus, this indeed remained way too implicit in our presentation. In order to address the reviewer’s criticism, we have now explicitly included this alternative and explain why it cannot account for the data. We chose (Shannon) information maximisation (infomax) as a relevant alternative because it is a widely investigated and accepted optimisation goal for sensory systems (Dayan and Abbott, 2000; Rieke et al., 1997, Spike: Exploring the Neural Code). We have now formulated infomax in the context of our network and built on a well-known relationship between the performance of the Bayesian decoder and mutual information, the objective function for infomax (Materials and methods). We have included an analysis of information transmission in the context of changing firing thresholds under phase nuisance parameter uncertainty in the Results section, included an additional panel on this analysis (Figure 3—figure supplement 2B), expanded the analysis of experimental data to the infomax principle, and discuss this alternative hypothesis in the Discussion. We have included the description of the infomax model in the Materials and methods. In short, our results indicate that infomax does not account for the experimentally found firing thresholds of V1 cells.

As for robustness to the choice of a specific RU-related objective, we have conducted another set of novel analyses to investigate RU performance when the goal was decoding spatial period or contrast rather than orientation. We performed the experiment by keeping other nuisance parameters unknown but swapping spatial period (or contrast) and orientation: orientation became a nuisance while spatial period (or contrast) became the variable of interest (Figure 4—figure supplement 2). Our analyses showed that while there were quantitative differences in performance from the original (orientation decoding) case, our main results carried over to these scenarios: the firing rate nonlinearity could increase (linear) decodability, and there was an optimal value for the threshold of this nonlinearity very near (within 1-2 mV) of the optimal threshold we originally found. A new figure which summarises these results has been added to the Appendix 1 (Figure 4—figure supplement 2). We refer to these results in the Results, discuss them in the Discussion, and have extended the Materials and methods with the description of this alternative decoding paradigm.

Similarly, the authors focused solely on RU of the orientation of sinusoidal gratings from the responses of model simple cells. Optimality of the firing rate threshold is defined in the context of untangling information about the orientation of sinusoidal gratings, absent any reference to why posing the problem this way makes sense, given that these neurons presumably evolved to process natural images.

While we agree that the evolutionary goal of the system must be to achieve maximal performance on natural images, rather than the highly simplified full field gratings we used in our study, our choice for these stimuli was motivated by a number of factors. First, as a large swathe of the theoretical and experimental literature has used such full-field grating stimuli, we thought it was important to use a paradigm which was as similar to previous work as possible in order to make our results easily comparable to those previous results. Indeed, we think that the point that nuisance parameters, rather than single-neuron variability or the exact structure of noise correlations, are the main bottleneck for decoding (and that this bottleneck is at least partially alleviated by the firing rate nonlinearity) is stronger when we show that it is even true for the same simple stimuli that others have used (but largely ignored the effects of nuisance parameters), even without considering the whole complexity of natural images.

Second, the relatively simple feed-forward architecture of our model population is probably sufficient and has been widely used to account for responses to such simple stimuli. However, more complex stimuli will recruit mechanisms (eg. those responsible for extra-classical receptive field effects) that this architecture cannot capture, we leave the effects of these mechanisms on RU for later studies. We now dedicate a section in the Discussion to cover this point (“Visual processing in ecologically relevant regimes”). In addition, as we explain it in the text, natural image statistics were at least taken into account in the choice of the distribution of nuisance parameters (Figure 5A).

Third, as we now show in Figure 3—figure supplement 3 (and discuss in the Results), our main results remain essentially unchanged when we consider a “hypercolumnar” population representing *local* rather than *global* orientation (ie. such that all cells have their receptive fields in the same location). Importantly, inasmuch as our model represents an appropriate approximation of hypercolumnar responses (see above), the content of natural images outside this (classical) receptive field location will not affect the decoding of the content at this location, and thus these results should generalise to natural stimuli.

2) Try to strengthen the presented evidence for RU from the image input to this level of V1 (spiking outputs of V1 simple cells).Previous experimental results advocating for RU along the ventral stream have made comparisons at two points of processing (e.g. V4 population spike rates and IT population spike rates). It would strengthen the claims if the authors would do something similar across the key stage of processing they are focused on here. E.g. compare with pixels, center-surround (e.g. difference of gaussians (DOG) to approx. RGC and LGN responses).We understand that some decodes were attempted on the simulated intracellular potential population and compared with the spiking outputs. That is good, but this seems a bit like comparing apples and oranges.

Thank you for this other excellent suggestion. We have taken it on board and now devote a whole new section (Representational untangling of orientation is specific to V1 and a section in the Appendix 1 (Section 1) and corresponding section in the Materials and methods) and new figure panels (Figure 3C,D) to showing that RU of orientation information cannot be achieved with retinal- (ie. single-pixel) or LGN-like (ie. DoG) receptive fields.

3) Better define the RU hypothesis (or alternative variants of it). In particular, the operational definition of linear separability is not agreed upon in the field, and different definitions tend to test different things. […]To summarize: the improved linear decodabilty of the V1 population relative to a pixel or DOG representation should be shown directly and the dependence on training sample size should be explored in some way. The key result might be: for limited number of training examples, the properly thresholded V1 decoding performance is higher than the pixel and DOG representations, but that result is not true in the limit of infinite training data. To be clear, either of the possibilities above are interesting and publishable, but they are impossible to disentangle (;) right now.

We trained our linear decoders with sufficient amounts of data so that to achieve asymptotic performance. This was true for *all* cases, including those using pixel or DoG (center-surround) receptive fields. (Thus, as the reviewer suggests, we mostly only test generalisation performance for new noise samples, not new stimuli, but see below.) The reason for this was twofold. First, we wanted to be able to make what we believe is the stronger statement: that *even in the limit of infinite training data* RU is non-trivial in the presence of nuisance parameters (and that the appropriate firing rate nonlinearity can help with it). Second, as we are studying the representation of low-level visual features in an early visual area, there is good reason to believe that the decoding of this information could have been optimised on evolutionary time scales (as the reviewer also notes elsewhere) and would thus not be limited by the amount of data experienced over the lifetime of an individual.

We have now clarified what we mean by generalisation in the Results, discuss our choice of asymptotic performance in the Discussion, and explain the details of cross validation in the Materials and methods and Table 1.

Furthermore, note that in Figure 5, we test a specific (and we believe ecologically relevant) form of generalisation: generalisation to new *distributions* of stimuli(rather than to new *individual* stimuli) which can occur when transitioning between different environments. Nevertheless, we also tested cross-validated performance with phase as nuisance parameter such that the distribution from which training and test stimuli were sampled were the same but the actual set of training and test stimuli were different and found that the performance vs. threshold curves (not shown) were near-identical to those we obtained originally (Figure 3).

Related: an intuitive solution to increase linearly separability is to set RELU thresholds very high such that the representation of each image is ultra-sparse. In the limit, each image would be coded as one-hot population response vector. Under some operational definitions of linear separability, this solution is perfectly fine (e.g. test only at the training images). However, it is not a real world solution because it has little to no generalization power and it requires a very large number of neurons to maintain information and corresponding large numbers of training examples. Thus, the trick for real world tests of linear separability in the setting explored in this paper is to set the thresholds high enough to enable linear separability in the context of limited training data and test generalization to new nuisance values of the stimulus parameters – what that limit should be and how "far" the "new" stimuli should be are admittedly unclear, but both could be considered and explored to firm up the arguments. Perhaps this has already been done and it was missed it in the presentation.

This is an intriguing possibility, thank you for bringing it up. As the reviewer anticipates, the one-hot population coding solution achieved by ultra-sparse codes fundamentally relies on assuming no noise, and requires an exponential number of neurons (in the number of nuisance parameters). In fact, in the (unrealistic) limit of no noise, the question of an optimal threshold even becomes somewhat moot as essentially *all* thresholds above a minimum will perform equally well (essentially perfectly), as the broadening of the robust performance interval towards low values of the noise-to-signal ratio in Figure 7C (and Figure 7—figure supplement 3) also suggests. Moreover, ultra sparse coding will be particularly sensitive to contrast as a nuisance parameter (as the correct threshold for achieving a one-hot code will critically depend on the overall scaling of responses which in turn depends monotonically on contrast, such that selecting a single optimal thresholdis impossible). All-in-all, ultra sparse codes were not relevant for our study because these conditions (especially no noise) were not met in our simulations. Indeed, note that we found that increasing the number of neurons in the population did not favour higher thresholds which could have potentially led to such ultra sparse codes (Figure 6B). More importantly, we expect these conditions also not to apply to real V1, e.g. the experimentally measured levels of noise-to-signal ratio in our V1 data were well above 0 (Figure 7). In sum, the pathological solution of ultra-sparse codes is already ruled out under the realistic conditions we studied. Note that this did not even require us to measure generalisation to new stimuli (but see Figure 5 and our previous response), only to new noise samples. (See our previous response for the justification of why we focussed on this form of generalisation.) We now discuss this in the text (subsection “The computational role of the FRNL”).